# CONDITIONAL INFORMATION BOTTLENECK APPROACH FOR TIME SERIES IMPUTATION

**MinGyu Choi**
Massachusetts Institute of Technology, USA
chemgyu@mit.edu

**Changhee Lee**
Chung-Ang University, Korea
changheelee@cau.ac.kr

## ABSTRACT

Time series imputation presents a significant challenge because it requires capturing the underlying temporal dynamics from partially observed time series data. Among the recent successes of imputation methods based on generative models, the *information bottleneck* (IB) framework offers a well-suited theoretical foundation for multiple imputations, allowing us to account for the uncertainty associated with the imputed values. However, direct application of IB framework to time series data without considering their temporal context can lead to a substantial loss of temporal dependencies, which, in turn, can degrade the overall imputation performance. To address such a challenge, we propose a novel *conditional information bottleneck (CIB)* approach for time series imputation, which aims to mitigate the potentially negative consequences of the regularization constraint by focusing on reducing the redundant information conditioned on the temporal context. We provide a theoretical analysis of its effect by adapting variational decomposition. We use the resulting insight and propose a novel deep learning method that can approximately achieve the proposed CIB objective for time series imputation as a combination of evidence lower bound and novel temporal kernel-enhanced contrastive optimization. Our experiments, conducted on multiple real-world datasets, consistently demonstrate that our method significantly improves imputation performance (including both interpolation and extrapolation), and also enhances prediction performance based on the imputed values.

## 1 INTRODUCTION

Multivariate time series data often includes missing features, with diverse missing ratios and patterns depending on distinct sampling periods or measurement strategies (Johnson et al., 2016). Since these missing features can significantly impair the performance of downstream tasks and comprehension of the temporal dynamics, time series imputation, which aims to reconstruct the missing features, has become a pivotal and pervasive topic across numerous practical domains, including healthcare, environmental science, and various other fields. What makes time series imputation challenging is that an imputation method must satisfy two essential requirements: i) it must account for underlying temporal dependencies, and ii) it should allow for *multiple imputations* to facilitate uncertainty quantification for real-world decision-making.

Generative models, particularly variational autoencoders (VAEs) (Kingma & Welling, 2014), have been employed in the context of multiple imputation tasks due to their capability to generate samples in a probabilistic manner. VAE-based imputation methods primarily focus on defining the evidence lower bound, where the reconstruction error is computed only over the observed part of the incomplete data (Sohn et al., 2015; Nazabal et al., 2020). These methods can be naturally interpreted under the information bottleneck (IB) principle (Tishby & Zaslavsky, 2015), providing an information-theoretic understanding of what constitutes an imputation-relevant representation. This understanding is based on the fundamental trade-off between maintaining a concise representation (i.e., regularization) and preserving good representation power (i.e., reconstruction) (Voloshynovskiy et al., 2019).

However, a direct application of the IB principle to time series imputation struggles with capturing the underlying temporal dependencies. Our motivating examples in Figure 1(B) show that imputation methods trained with the conventional IB framework lose a significant amount of information about temporal dynamics relevant for imputing missing values. In this paper, we theoretically analyze that

the overly strict regularization in the conventional IB may force the encoder to rely solely on the observed features at a particular time point, rather than learning the underlying temporal dependencies from the remaining observations from other time steps. To overcome such an issue, we propose a novel *conditional information bottleneck* (CIB) framework for time series imputation. Our framework adopts the reconstruction-regularization structure of the IB principle while preserving temporal information through conditional regularization, allowing us to circumvent the strict regularization constraints of the conventional IB. Throughout the experiments conducted on multiple real-world datasets including image sequences, weather measurements, and electrical health records, our proposed method consistently outperforms the state-of-the-art imputation methods with respect to both imputation performance and prediction performance based on the imputed values.

## 2 PRELIMINARIES: INFORMATION BOTTLENECK APPROACH TO IMPUTATION

In this section, we first formally describe the information bottleneck (IB) principle (Tishby & Zaslavsky, 2015; Alemi et al., 2017), which provides an information-theoretic understanding of what a task-relevant representation is in terms of the fundamental trade-off between having a concise representation and good representative power. Then, we present a generative model for imputing missing features under the IB principle.

Let $\mathbf{X}$ and $\mathbf{Y}$ be random variables for the input feature and the target label, respectively. The IB principle aims to find the bottleneck random variable $\mathbf{Z}$ that compresses the information in $\mathbf{X}$ while keeping the information relevant for predicting $\mathbf{Y}$ as the following (Tishby & Zaslavsky, 2015),

$$\min_{\phi,\theta} \ I_\phi(\mathbf{Z};\mathbf{X}) - \beta I_\theta(\mathbf{Y};\mathbf{Z}) \tag{1}$$

where $\beta \in \mathbb{R}$ is a Lagrangian multiplier that balances the two mutual information terms, and $\phi$ and $\theta$ correspond to learnable parameters that define probabilistic mappings $q_\phi(\mathbf{Z}|\mathbf{X})$ and $q_\theta(\mathbf{Y}|\mathbf{Z})$, respectively. The core motivation of (1) is to find the optimal distribution of latent representation $\mathbf{Z}$ and the corresponding inference model parameters $\phi$ that removes label-irrelevant information from $\mathbf{X}$ while preserving the information about the class label $\mathbf{Y}$.

This offers an information-theoretic perspective on generative model-based imputation methods which generate missing observations from the observed features.

**Definition 1.** *(Imputation) Let $\mathbf{X}^o$ and $\mathbf{X}^m$ be random variables for the partially observed features and missing features of $\mathbf{X}$, respectively, such that $\mathbf{X} = \mathbf{X}^o \cup \mathbf{X}^m$. Then, we define imputation as an unsupervised IB as follows:*

$$\min_{\phi,\theta} \ I_\phi(\mathbf{Z};\mathbf{X}^o) - \beta I_\theta(\mathbf{X};\mathbf{Z}) \tag{2}$$

*where $\beta \in \mathbb{R}_\geq$ is a Lagrangian multiplier, and $\phi$ and $\theta$ correspond to learnable parameters that define probabilistic mappings $q_\phi(\mathbf{Z}|\mathbf{X}^o)$ and $q_\theta(\mathbf{X}|\mathbf{Z})$, respectively.*

The above definition in (2) aims at finding the distribution of latent representation $\mathbf{Z}$ and the corresponding parameters $\phi$ that preserves the core information for accurately reconstructing the (complete) original input $\mathbf{X}$ while suppressing redundant information from its incomplete observation, $\mathbf{X}^o$.

## 3 METHOD

### 3.1 PROBLEM FORMULATION

We consider a general temporal dynamics setting in which each instance of a (discrete) time series input comprises a sequence of measurements (i.e., observations), denoted as $\mathbf{x}_{1:T} \stackrel{\text{def}}{=} [\mathbf{x}_1, \dots, \mathbf{x}_T]$, collected during the time interval $[0, \tau_T]$. Here, $\mathbf{x}_t \in \mathbb{R}^d$ is the complete input vector measured at time $t \in [\tau_{t-1}, \tau_t)$ and is a realization of a random variable $\mathbf{X}_t$.[1] However, in practice, time series data often contains missing features with arbitrary patterns such that $x_t^l$ is not observed during $[\tau_{t-1}, \tau_t)$ for any feature $l \in \{1, \dots, d\}$ at any time step $t \in \{1, \dots, T\}$. This phenomenon is particularly common in domains such as healthcare (Johnson et al., 2016) where each feature may

---

[1]Throughout the paper, we will often use upper-case letters to represent random variables and lower-case letters to represent their corresponding realizations. Please refer to Appendix E for a notation table.

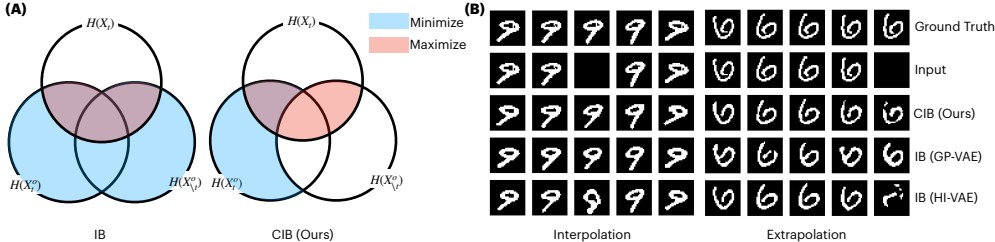

Figure 1: (A) Conceptual illustration of the IB and CIB principles. By conditioning regularization on the remaining input time steps, the latent representation can better preserve the underlying temporal dependency. (B) Motivating experimental results on interpolation (left) and extrapolation (right). Because features in a single time step are completely missing, a model must collect information from other time steps. The conventional IB approach (HI-VAE) shows deteriorating performance in both cases. Another IB approach (GP-VAE) using a Gaussian process prior demonstrates enhanced performance for interpolation but often significantly loses time series characteristics for extrapolation (i.e., the writing style is corrupted). The CIB approach (Ours) exhibits improved imputation performance for both cases. Complete quantitative results are available in Table 1.

have a distinct sampling period or when non-uniform sampling strategies are employed. To denote missing observations, we partition the input vector $\mathbf{x}_t$ at each time step into observed features $\mathbf{x}_t^o$ and missing features $\mathbf{x}_t^m$, such that $\mathbf{x}_t = \mathbf{x}_t^o \cup \mathbf{x}_t^m$.

**Objective.** Our aim is to reconstruct the complete time series input $\mathbf{x}_{1:T}$ by filling in the missing features from the observed time series input $\mathbf{x}_{1:T}^o$. Formally, we seek to generate $\mathbf{x}_t^m$ from the conditional distribution $p(\mathbf{X}_t^m | \mathbf{X}_{1:T}^o)$. By modeling the conditional distribution $p(\mathbf{X}_t^m | \mathbf{X}_{1:T}^o)$ instead of using a deterministic mapping, we can generate *multiple imputations*, allowing us to capture the uncertainty associated with the imputed values.

What makes this problem challenging is that we must account for the underlying temporal dynamics represented by $\mathbf{x}_{1:T}^o$ when imputing missing features $\mathbf{x}_t^m$ for $t \in \{1, \ldots, T\}$. We can straightforwardly apply the unsupervised IB described in (2) to obtain latent representations $\mathbf{Z}_t$ by discarding information from the observed time series input $\mathbf{X}_{1:T}^o$ that is redundant for reconstructing $\mathbf{X}_t$. Formally, this can be achieved by minimizing $I_\phi(\mathbf{Z}_t; \mathbf{X}_{1:T}^o) - \beta I_\theta(\mathbf{X}_t; \mathbf{Z}_t)$ with a comprehensive encoder (e.g., RNN or Transformer) capable of effectively modeling the temporal dependencies within the observed time series observations, i.e., $q_\phi(\mathbf{Z}_t | \mathbf{X}_{1:T}^o)$. However, enforcing such strict regularization constraints on the encoder may lead to a significant loss of information regarding the temporal context that can be achieved by observations at different time steps, which we denote as $\mathbf{X}_{\setminus t}^o \stackrel{\text{def}}{=} (\mathbf{X}_\tau^o : \tau \in \{1, \ldots, T\} \setminus t)$. This may cause the imputation of $\mathbf{X}_t^m$ at time step $t$ to heavily rely on the observed features at that particular time point, i.e., $\mathbf{X}_t^o$, rather than being able to learn from temporal dependencies present in other observations, i.e., $\mathbf{X}_{\setminus t}^o$ (as shown in Figure 1(B)).

To tackle this issue, we alleviate the potentially negative consequences of the regularization constraint by directing our attention to the redundant information of the observed input at time step $t$ when it is conditioned on its temporal context represented by the remaining observed time series $\mathbf{X}_{\setminus t}^o$. This offers a novel information-theoretic rationale for time series imputation, as defined below:

**Definition 2.** *(Time Series Imputation)* Let $\mathbf{X}_t^o$ and $\mathbf{X}_t^m$ be random variables for the partially observed features and missing features of $\mathbf{X}_t$ at time step $t$. Then, given the observed time series input $\mathbf{X}_{1:T}^o$, we define time series imputation at time step $t$ as an unsupervised CIB as follows:

$$\min_{\phi, \theta} \underbrace{I_\phi(\mathbf{Z}_t; \mathbf{X}_t^o | \mathbf{X}_{\setminus t}^o)}_{\textit{Conditional Regularization}} - \underbrace{\beta I_\theta(\mathbf{X}_t; \mathbf{Z}_t)}_{\textit{Reconstruction}} \tag{3}$$

*where* $\mathbf{X}_{\setminus t}^o$ *represents the random variables for the remaining input observations, excluding* $\mathbf{X}_t^o$.

By conditioning on $\mathbf{X}_{\setminus t}^o$, (3) guides us to find latent representations $\mathbf{Z}_t$ and the corresponding inference model parameter $\phi$ which encompass all retrievable information from the entire observed input time series $\mathbf{X}_{1:T}^o$ (*reconstruction*), while discarding information that is redundant for capturing $\mathbf{X}_t^m$ given the available temporal context from the remaining observed time series $\mathbf{X}_{\setminus t}^o$ (***conditional regularization***). Overall, the proposed objective in (3) enables us to more effectively utilize information

from $\mathbf{X}_{\setminus t}^o$ for imputing $\mathbf{X}_t^m$ compared to other IB-related alternatives, whose conceptual illustration can be seen in Figure 1(A).

## 3.2 DEEP VARIATIONAL CONDITIONAL INFORMATION BOTTLENECK ON TIME SERIES

In this subsection, we will transform our objective (3) into a learnable form by utilizing variational decomposition. (3) is represented as a combination of the traditional ELBO with mutual information along the time axis, which can be approximately achieved by minimizing the contrastive loss.

### 3.2.1 MAXIMIZING RECONSTRUCTION: $\min_{\phi,\theta} -I(\mathbf{X}_t; \mathbf{Z}_t)$

Following the derivations introduced in (Voloshynovskiy et al., 2019), we can find a lower bound of the reconstruction term as the following:

$$
I_\theta(\mathbf{X}_t; \mathbf{Z}_t) = \mathcal{H}(\mathbf{x}_t) + D_{\mathrm{KL}}(p(\mathbf{x}_t|\mathbf{z}_t)||p_\theta(\mathbf{x}_t|\mathbf{z}_t)) + \mathbb{E}_{\mathbf{x}_{1:T}^o \sim p_{\mathrm{data}}} \left[ \mathbb{E}_{\mathbf{z}_t \sim q_\phi(\mathbf{z}_t|\mathbf{x}_{1:T}^o)} \left[ \log p_\theta(\mathbf{x}_t|\mathbf{z}_t) \right] \right]
$$

$$
\geq \mathbb{E}_{\mathbf{x}_{1:T}^o \sim p_{\mathrm{data}}} \left[ \mathbb{E}_{\mathbf{z}_t \sim q_\phi(\mathbf{z}_t|\mathbf{x}_{1:T}^o)} \left[ \log p_\theta(\mathbf{x}_t|\mathbf{z}_t) \right] \right] \stackrel{\mathrm{def}}{=} -\mathcal{L}_{\phi,\theta}^1 \tag{4}
$$

where $\mathcal{H}(\cdot)$ is the entropy and the last inequality holds due to the non-negativity of entropy and KL-divergence. Here, we introduce a *feature estimator*, denoted as $p_\theta(\mathbf{X}_t|\mathbf{Z}_t)$, as a variational approximation of $p(\mathbf{X}_t|\mathbf{Z}_t)$. We model the feature estimator as an isotropic Gaussian, i.e., $p_\theta(\mathbf{X}_t|\mathbf{Z}_t) = \mathcal{N}(\mu_\theta(\mathbf{Z}_t), \mathrm{diag}(\sigma_\theta(\mathbf{Z}_t)))$ where $\mu_\theta(\cdot)$ and $\sigma_\theta(\cdot)$ are implemented by neural networks parameterized by $\theta$.

In many practical scenarios, the ground-truth values for missing features are unknown during training. Thus, to accurately learn the reconstruction process given the latent representation of the observed time series, we apply (4) only to the features observed at each time point, similar to the approach in (Nazabal et al., 2020).

**Discussion on the Conditional Reconstruction.** One might question why the reconstruction term is not conditioned on $\mathbf{X}_{\setminus t}^o$, as given by an alternative form of the CIB, i.e., $\min_{\phi,\theta} I_\phi(\mathbf{Z}_t; \mathbf{X}_t^o|\mathbf{X}_{\setminus t}^o) - \beta I(\mathbf{X}_t; \mathbf{Z}_t|\mathbf{X}_{\setminus t}^o)$. Applying the chain rule of mutual information[2] decomposes the conditional reconstruction as the follows: $I(\mathbf{X}_t; \mathbf{Z}_t|\mathbf{X}_{\setminus t}^o) = I(\mathbf{X}_t; \mathbf{Z}_t, \mathbf{X}_{\setminus t}^o) - I(\mathbf{X}_t; \mathbf{X}_{\setminus t}^o)$. It turns out that the first term can be bounded by a mathematically equivalent expression as shown in (4), suggesting that this term encourages mitigating constraints on temporal context for reconstruction (see Appendix A.1). However, minimizing the second term attempts to eliminate information about the target $\mathbf{X}_t$ at time point $t$, which can be achieved from the observation other than time point $t$, i.e., $\mathbf{X}_{\setminus t}$. This contradicts the goal of time series imputation where we aim to capture temporal context from the remaining observed time steps. Our empirical results also support that minimizing $I(\mathbf{X}_t; \mathbf{X}_{\setminus t}^o)$ deteriorates the model performance (see Appendix A.2 for derivation, B.2 for experimental results).[3]

### 3.2.2 MINIMIZING CONDITIONAL REGULARIZATION: $\min_{\phi,\theta} I_\phi(\mathbf{Z}_t; \mathbf{X}_t^o|\mathbf{X}_{\setminus t}^o)$.

We employ the chain rule for mutual information on the conditional regularization term as follows:

$$
\min_{\phi,\theta} I(\mathbf{Z}_t; \mathbf{X}_t^o|\mathbf{X}_{\setminus t}^o) = \min_{\phi,\theta} I(\mathbf{Z}_t; \mathbf{X}_{1:T}^o) - I(\mathbf{Z}_t; \mathbf{X}_{\setminus t}^o). \tag{5}
$$

It is worth highlighting that the application of the chain rule decomposes the conditional regularization into two components: (i) minimizing the information between the latent representation $\mathbf{Z}_t$ and the entire observed time series input $\mathbf{X}_{1:T}^o$ that encourages the latent representation to be concise, while (ii) maximizing the information from $\mathbf{X}_{\setminus t}^o$ to capture the underlying temporal dynamics provided by the observations at the remaining time steps. This prevents a significant loss of temporal context in the IB and, in turn, enhances the utilization of temporal dependencies from the remaining time steps.

**Minimizing $I(\mathbf{Z}_t; \mathbf{X}_{1:T}^o)$.** The first term in (5) can be bounded as follows (see Appendix A.3):

$$
I(\mathbf{Z}_t; \mathbf{X}_{1:T}^o) \leq \mathbb{E}_{\mathbf{x}_{1:T}^o \sim p_{\mathrm{data}}} [D_{\mathrm{KL}}(q_\phi(\mathbf{z}_t|\mathbf{x}_{1:T}^o)||p(\mathbf{z}_t))] \stackrel{\mathrm{def}}{=} \mathcal{L}_\phi^2 \tag{6}
$$

---

[2]Let $\mathbf{V}$, $\mathbf{W}$, and $\mathbf{Y}$ be random variables, then the chain rule gives $I(\mathbf{Y}; \mathbf{W}|\mathbf{V}) = I(\mathbf{Y}; \mathbf{W}, \mathbf{V}) - I(\mathbf{Y}; \mathbf{V})$.

[3]Conditional reconstruction can be appropriate for capturing information that exclusively depends on the corresponding input, as introduced in (Fischer, 2020; Lee et al., 2023).

where we utilize the unit isotropic Gaussian as the prior distribution, i.e., $p(\mathbf{Z}_t) = \mathcal{N}(\mathbf{0}, I)$. We model the *stochastic encoder* as a multivariate Gaussian distribution defined as $q_\phi(\mathbf{Z}_t|\mathbf{X}_{1:T}^o) = \mathcal{N}(\mu_\phi(\mathbf{X}_{1:T}^o), \text{diag}(\sigma_\phi(\mathbf{X}_{1:T}^o)))$, where $\mu_\phi(\cdot)$ and $\sigma_\phi(\cdot)$ are implemented as neural networks parameterized by $\phi$. This explains why the (unconditional) IB struggles with modeling temporal dynamics, as discussed in Section 3.1. That is, (6) forces the encoder mappings that depend on time series inputs to converge to the unit Gaussian, imposing overly strict regularization. This hinders capturing of temporal dependencies present in other observations, motivating us to explicitly capture temporal dynamics by maximizing $I(\mathbf{Z}_t; \mathbf{X}_{\backslash t}^o)$ rather than relying solely on the reconstruction signal in (4).

**Maximizing $I(\mathbf{Z}_t; \mathbf{X}_{\backslash t}^o)$.** To bound the second term in (5), we adopt the InfoNCE minimization from the contrastive learning on latent representations that approximately achieves maximizing the corresponding mutual information (Oord et al., 2018; Tian et al., 2020). Suppose we have the latent representation at time step $t$ given an observed time series $\mathbf{x}_{1:T}^o$ as a reference, i.e., $\mathbf{z}_t \sim q_\phi(\mathbf{Z}_t|\mathbf{x}_{1:T}^o)$. Since our aim is to maximize information from $\mathbf{X}_{\backslash t}^o$, we intentionally make missing observations at time step $t$ from the reference, and employ latent representations of time steps other than $t$ as positive pairs. For negative pairs, we consider latent representations at different time steps from other time series observations. Finally, we define our novel contrastive learning loss with cosine similarity of latent representations along the time axis as follows (see Appendix A.4):

$$I(\mathbf{Z}_t; \mathbf{X}_{\backslash t}^o) \geq \mathbb{E}_{\mathbf{x}_{1:T}^o \sim p_{\text{data}}} \left[ \log \left( \frac{\sum\limits_{t' \in \{1,...,T\} \backslash t} \exp\left(\mathbf{z}_t^T \tilde{\mathbf{z}}_{t'}/\tau\right)}{\sum\limits_{\substack{\mathbf{x}_{1:T}^- \in \mathcal{X}_{1:T}^-}} \sum\limits_{\substack{t' \in \{1,...,T\}, \\ \mathbf{z}_{t'}^- \sim q_\phi(\mathbf{z}_{t'}^-|\mathbf{x}_{1:T}^-)}} \exp\left(\mathbf{z}_t^T \mathbf{z}_{t'}^-/\tau\right)} \right) \right] \overset{\text{def}}{=} -\mathcal{L}_\phi^3 \quad (7)$$

where $\tau$ is the temperature parameter. Here, $\tilde{\mathbf{z}}_{t'} \sim q_\phi(\tilde{\mathbf{Z}}_{t'}|\mathbf{x}_{\backslash t}^o)$ denotes the positive pair obtained by masking the reference time series, such that $\mathbf{x}_{\backslash t}^o$ is created by replacing $\mathbf{x}_t^o$ with zeros from $\mathbf{x}_{1:T}^o$. We regard such positive pairs as augmentations of a given time series since latent representations with missing values at time step $t$ share task-relevant information about the underlying temporal dynamics of a given time series. We denote $\mathcal{X}_{1:T}^-$ a set of negative samples comprising other time series in the same mini-batch, where $\mathbf{x}_{1:T}^-$ indicates an observed time series from $\mathcal{X}_{1:T}^-$. This makes our encoder capture time series-level semantics – such as underlying disease progression patterns that can be distinguished from others – by pushing these samples from the reference. Such an attribution is necessary for reconstructing missing values (and associated downstream tasks in the experiments) specific to the input time series. Please refer to Appendix C for implementation details.

### 3.2.3 OPTIMIZATION

Now, we introduce a novel imputation method, which we refer to as *Time-series Imputation using Conditional Information Bottleneck (**TimeCIB**)*, that consists of the stochastic encoder, $q_\phi$, and the feature estimator, $p_\theta$, introduced above. Please see Figure C1 for a schematic illustration of our framework. Overall, we optimize our method based on the following objective by combining all the loss functions that allows us to approximately achieve time series imputation defined in (3):

$$\min_{\phi,\theta} \quad \beta\mathcal{L}_{\phi,\theta}^1 + \mathcal{L}_\phi^2 + \gamma\mathcal{L}_\phi^3 \quad (8)$$

where $\gamma \in \mathbb{R}_{\geq 0}$ is a balancing coefficient that trades off the impact of $\mathcal{L}_\phi^3$. We provide sensitivity analysis on $\beta, \gamma$ in Appendix B.4.

### 3.3 INTRODUCING INDUCTIVE BIAS ABOUT TEMPORAL DYNAMICS

Now, we illustrate how we can inject inductive bias about the underlying temporal dynamics by employing temporal kernels to further improve the expressive power of TimeCIB.

The *alignment* of the latent representation (Wang & Isola, 2020), attained through contrastive learning based on (7), renders the similarity between latent representations at two adjacent time points indistinguishable from the similarity between those at two distant time points. This phenomenon appears to contradict real-world temporal dynamics, such as gradually deteriorating or periodic behavior of disease progression patterns. To address this, we employ *conditional alignment* (Dufumier et al., 2021) that introduces inductive bias about the underlying temporal dynamics with temporal kernels as the following:

$$I(\mathbf{Z}_t; \mathbf{X}^o_{\backslash t}) \geq \mathbb{E}_{\mathbf{x}^o_{1:T} \sim p_{\text{data}}} \left[ \log \left( \frac{\sum\limits_{t' \in \{1,\ldots,T\}\backslash t} c_{t,t'} \exp\left(\mathbf{z}_t^T \tilde{\mathbf{z}}_{t'}/\tau\right)}{\sum\limits_{\mathbf{x}^-_{1:T} \in \mathcal{X}^-_{1:T}} \sum\limits_{\substack{t' \in \{1,\ldots,T\}, \\ \mathbf{z}^-_{t'} \sim q_\phi(\mathbf{z}^-_{t'}|\mathbf{x}^-_{1:T})}} \exp\left(\mathbf{z}_t^T \mathbf{z}^-_{t'}/\tau\right)} \right) \right] \overset{\text{def}}{=} -\mathcal{L}^{3'}_\phi \quad (9)$$

where $c_{t,t'} \in \mathbb{R}$ is a kernel coefficient as a function of two time points $t$ and $t'$.

Incorporating prior knowledge of underlying similarity into contrastive learning is not a novel concept. Several previous works have leveraged supervisory information to adapt contrastive learning, including semantic similarities for text classification (Suresh & Ong, 2021), angle similarity for gaze estimation (Wang et al., 2022), and Gaussian priors for time series and video representation learning (Tonekaboni et al., 2021; Chen et al., 2022). In this paper, we evaluate the following two representative temporal kernels to evaluate our method on data with different temporal behaviors and will leave the choice of the kernel as a hyperparameter:

$$c_{\text{cauchy}}(\tau, \tau') = \sigma^2 \left(1 + \frac{(\tau - \tau')^2}{l^2}\right)^{-1}, \quad c_{\text{periodic}}(\tau, \tau') = \sigma^2 \exp\left(-\frac{2\sin^2\left(\pi(\tau - \tau')/p\right)}{l^2}\right) \quad (10)$$

- **Cauchy Smoothing.** Under the assumption that two nearby time points should be more similar than those far away, we smooth the latent representations of time series by assigning higher weights to nearby time points when pulling the representations, utilizing the *Cauchy kernel* defined as $c_{\text{cauchy}}(\tau, \tau')$ in (10) which is the mixture of infinite RBF kernels with different time scales (Rasmussen, 2004). This is a generalized form that reduces to uniform weights as in (7), i.e., $l = \infty$ gives $c(\tau, \tau') = \sigma^2$.
- **Periodic Smoothing.** Unfortunately, Cauchy smoothing may not be appropriate when the underlying temporal dynamics exhibit periodic behavior (e.g., seasonality). To incorporate our domain knowledge about periodic time series data, we utilize the exponentiated sine-squared kernel given as $c_{\text{periodic}}(\tau, \tau')$ in (10) where $l \in \mathbb{R}$ corresponds to the length scale and $p \in \mathbb{R}$ reflects the periodicity.

## 4 RELATED WORKS

**Time Series Imputation: Predictive Methods.** Earlier works on time series imputation have been proposed for *single imputation* utilizing predictive models. M-RNN (Yoon et al., 2018b) and BRITS (Cao et al., 2018) are representative RNN-based methods that predict missing observations by employing bidirectional RNNs to capture both past and future temporal dependencies. Inspired by the recent success of transformers in modeling time series data over conventional RNN architectures, recent predictive methods for time series imputation employ a self-attention mechanism to enhance imputation performance (Bansal et al., 2021; Du et al., 2023; Shan et al., 2023). However, these methods cannot provide multiple imputations and, therefore, fail to incorporate the uncertainty associated with imputed values.

**Time Series Imputation: Generative Methods.** Two main strands of generative models – VAEs (Kingma & Welling, 2014) and generative adversarial networks (GANs) (Makhzani et al., 2015) – have been introduced for multiple imputation due to their ability to *stochastically* generate samples. Here, we describe VAE-based methods. See Appendix D.3 for more related works.

VAE-based imputation methods primarily focus on defining the evidence lower bound, where the reconstruction error is computed only over the observed part of the incomplete data while missing values are filled with arbitrary values (e.g., zeros) during inference (Sohn et al., 2015; Nazabal et al., 2020). Fortuin et al. (2020) proposed GP-VAE which adopts a similar approach to efficiently handle incomplete (missing) data in a temporal setting by assuming that the latent representation of input time series evolves smoothly over time according to a Gaussian process (GP). While introducing the GP prior improves the ability to capture the underlying temporal dynamics, GP-VAE still cannot capture shared temporal structures across time series data, as it employs an independent GP prior for each time series. More recently, L-VAE (Ramchandran et al., 2021) and its conditional extension (Ramchandran et al., 2022) further improve the GP prior of GP-VAE by utilizing auxiliary covariates information. We focus our comparison on VAE-based models since these models can be information-theoretically interpreted as optimizing the IB(Voloshynovskiy et al., 2019).

It is worth clarifying that TimeCIB can be distinguished from GP-VAE in terms of how they achieve time series imputation and leverage temporal kernels. Under the IB framework, GP-VAE models the smooth temporal evolution of latent variables by replacing the traditional unit Gaussian prior with a GP prior specified by temporal kernels. However, TimeCIB is motivated by the inherent limitation of the IB in discarding temporal information (see Section 3.1) and proposes a novel CIB principle that alleviates the strict regularization of IB. Temporal kernels are optionally adopted to introduce an inductive bias to the underlying temporal dynamics.

**Information Bottleneck with Conditional Information.**   Several works have tailored the IB principle (Tishby & Zaslavsky, 2015; Alemi et al., 2017) by introducing conditional reconstruction or applying conditional regularization to extract information in alignment with their specific objectives. (Gondek & Hofmann, 2003) proposed conditional reconstruction to discover a new meaningful set of clusters that is orthogonal to the known class labels. (Fischer, 2020; Tezuka & Namekawa, 2021) introduced conditional regularization for supervised learning, which minimizes only redundant information given label information, thereby preventing the loss of label-related information due to overly strict regularization in the conventional IB principle. More recently, (Lee et al., 2023) utilized both conditional reconstruction and regularization to discover a label-related core subgraph from a pair of two molecular graphs.

From this perspective, our work aligns with conditional regularization approaches. While previous works have primarily focused on mitigating regularization concerning target label information, our method aims to alleviate overly strict regularization that can hinder the learning of underlying temporal dynamics. Moreover, to the best of our knowledge, this is the first work that presents an information-theoretic definition for time series imputation and proposes a novel conditional IB that can effectively preserve temporal dynamics for better imputation.

## 5  EXPERIMENTS

### 5.1  EXPERIMENTAL SETUP

**Evaluation Metrics.**   We evaluate the imputation performance from two perspectives: i) Imputation performance which measures feature-wise (pixel-wise) reconstruction. Specifically, we assess the negative log-likelihood (*NLL*) and mean squared error (*MSE*) of the imputed values on artificially missing features. ii) Prediction performance, which indirectly measures how well the imputed values preserve task-relevant information, which is a crucial aspect of imputation methods in practice. Following the experimental setup in (Fortuin et al., 2020) and (Yoon et al., 2019), we train separate classifiers or predictors with imputed values to predict the target labels. Then, we evaluate the area under the receiver operating characteristic (*AUROC*) for classification tasks and the MSE of the forecast (*ForecastMSE*) for forecasting tasks to measure the discriminative and predictive performance of imputation methods, respectively.

**Baseline Models.**   We focus our comparison on VAE-based models since these models can be interpreted under the IB principle as suggested in (Voloshynovskiy et al., 2019). Moreover, these multiple imputation methods can provide uncertainty of the imputed values, which is often crucial to support decision-making processes such as clinical interventions in healthcare. Hence, for baseline models, we compare our proposed method with the following: i) **GP-VAE** (Fortuin et al., 2020) which utilizes the Gaussian process (GP) prior to model time dependency, ii) **HI-VAE** (Nazabal et al., 2020) and iii) **VAE** (Kingma & Welling, 2014), both of which use an autoencoder architecture and are capable of imputing values at each time step. In addition to the above baselines, we compare with cutting-edge predictive methods: an RNN-based method, **BRITS** (Cao et al., 2018), and a transformer-based methods, **SAITS** (Du et al., 2023). Moreover, we also compare with state-of-the-art generative imputation methods, attention-based autoencoder approach, **mTANs** (Shukla & Marlin, 2021) , and diffusion-based approach, **CSDI** (Tashiro et al., 2021).[4] For a fair comparison, the magnitudes of the number of parameters are set to be equivalent among the evaluated methods. More detailed explanations about baseline models are provided in Appendix D.

---

[4]We compare NLL only with two VAE-based methods, since NLL cannot be measured for predictive methods; mTAN is probabilistic but uses fixed variance; and NLL of CSDI affected by the noise schedule. (please refer to Appendix F.2 of Tashiro et al. (2021)). We mark asterisks (*) in Table 2 and Table 3 to specify this.

Table 1: Imputation and prediction performance on the image sequence datasets.

| Methods | HealingMNIST (*missing with MNAR pattern*) | | | RotatedMNIST (*interpolation & extrapolation*) | |
|---|---|---|---|---|---|
| | NLL($\downarrow$) | MSE($\downarrow$) | AUROC($\uparrow$) | NLL($\downarrow$) | MSE($\downarrow$) |
| No Imp. | - | $0.293 \pm 0.000$ | $0.920 \pm 0.000$ | - | $0.133 \pm 0.000$ |
| Mean Imp. | - | $0.168 \pm 0.000$ | $0.938 \pm 0.000$ | - | $0.085 \pm 0.000$ |
| Forward Imp. | - | $0.177 \pm 0.000$ | $0.946 \pm 0.000$ | - | $0.080 \pm 0.000$ |
| VAE | $0.480 \pm 0.002$ | $0.232 \pm 0.000$ | $0.922 \pm 0.000$ | $1.773 \pm 0.127$ | $0.133 \pm 0.000$ |
| HI-VAE | $0.290 \pm 0.001$ | $0.134 \pm 0.003$ | $0.962 \pm 0.001$ | $0.207 \pm 0.007$ | $0.087 \pm 0.001$ |
| GP-VAE | $0.261 \pm 0.001$ | $0.114 \pm 0.002$ | $0.960 \pm 0.002$ | $0.190 \pm 0.001$ | $0.080 \pm 0.004$ |
| Ours(Uniform) | $0.204 \pm 0.002$ | $0.090 \pm 0.001$ | $\mathbf{0.967} \pm 0.001$ | $\mathbf{0.184} \pm 0.001$ | $0.077 \pm 0.001$ |
| Ours(Cauchy) | $\mathbf{0.202} \pm 0.004$ | $\mathbf{0.088} \pm 0.002$ | $\mathbf{0.967} \pm 0.000$ | $\mathbf{0.184} \pm 0.001$ | $\mathbf{0.076} \pm 0.002$ |

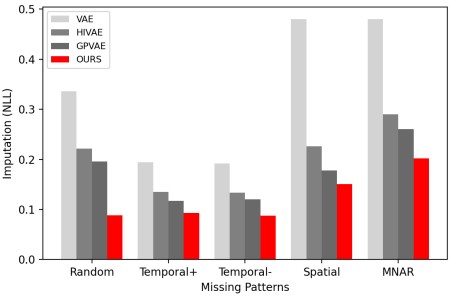

(a) Robustness on missing patterns.

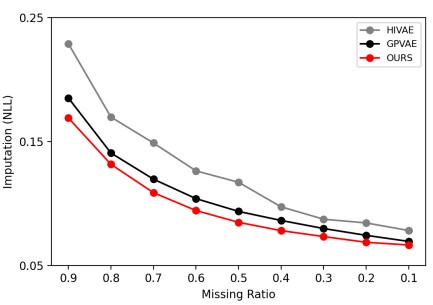

(b) Robustness on missing ratios.

Figure 2: Robustness analysis for missing patterns and missing ratios on HealingMNIST.

## 5.2 MAIN RESULTS

**Imputation on image sequences.** To evaluate imputation performance on diverse missing scenarios, we assess imputation performance on two MNIST sequence benchmarks with various missing patterns. HealingMNIST (Krishnan et al., 2015) has approximately $60\%$ of missing pixels under a missing-not-at-random (MNAR) pattern on every time step, where the missing probability of white pixels is twice larger than that of black pixels. Given that the model is not provided with information about the underlying missing mechanism, this task is particularly challenging, yet it mirrors many practical scenarios. For example, in healthcare, patients with depression are more likely to refuse answers about the severity of their condition (Gliklich et al., 2014). RotatedMNIST (Ramchandran et al., 2021) evaluates performance on interpolation and extrapolation, where all features at an arbitrary time step are completely missing. This makes imputation more challenging since the model must reconstruct all the missing values at a given time step solely based on the temporal dependency. Table 1 demonstrates that TimeCIB provides state-of-the-art imputation and prediction performance on both datasets, and the application of the Cauchy kernel (10) can further improve the performance.

We further evaluate the robustness of our model on HealingMNIST with four additional missing patterns with $60\%$ of missing ratio (Figure B2a) and that with different missing ratios ranging from $10\%$ to $90\%$ with the random missing pattern (Figure B2b). To assess the robustness of missing patterns, we employ *Random*, *Spatial* (i.e., neighboring pixels have correlated missing probabilities), and *Temporal+/−* (i.e., positive/negative temporal correlation). The imputation performance of our proposed method is most robust on diverse missing patterns and missing ratios. Please refer to Appendix B for more experimental details and results.

**Imputation for weather forecasting.** Weather forecasting is one of the representative fields where we can observe diverse scales of *seasonality* – such as daily, weekly, monthly, or yearly basis – which is one of the most important characteristics of time series data. In this experiment, we focused on two weather forecasting datasets - Beijing Air Quality (Zhang et al., 2017) and US Local[5] - whose time series measurements are collected every hour. Our model is capable of using the prior knowledge on the periodicity of the target data, by applying conditional alignment with temporal kernels (Section 3.3); we assume daily periodicity by setting $p = 24$. Inspired by the experiments outlined in (Yoon et al., 2019), we also evaluate the utility of imputation methods on forecasting by assessing the prediction performance (ForecastMSE) of a separately trained LSTM using time series data with imputed values. As shown in Table 2, TimeCIB outperforms the benchmarks on both weather forecasting datasets in terms of both imputation and prediction performance. The

---

[5]https://www.ncei.noaa.gov/data/local-climatological-data/

Table 2: Imputation and prediction performance on the weather forecasting datasets.

| Methods | Beijing (T=24) | | | US Local (T=168) | | |
|---|---|---|---|---|---|---|
| | NLL($\downarrow$) | MSE($\downarrow$) | ForecastMSE($\downarrow$) | NLL($\downarrow$) | MSE($\downarrow$) | ForecastMSE($\downarrow$) |
| No Imp. | - | $1.015 \pm 0.000$ | $0.539 \pm 0.000$ | - | $1.113 \pm 0.000$ | $0.610 \pm 0.000$ |
| Mean Imp. | - | $0.460 \pm 0.000$ | $0.517 \pm 0.000$ | - | $0.509 \pm 0.000$ | $0.432 \pm 0.000$ |
| Forward Imp. | - | $0.399 \pm 0.000$ | $0.502 \pm 0.000$ | - | $0.391 \pm 0.000$ | $0.401 \pm 0.000$ |
| BRITS | - | $0.396 \pm 0.002$ | $0.490 \pm 0.005$ | - | $0.384 \pm 0.001$ | $0.398 \pm 0.027$ |
| SAITS | - | $\mathbf{0.283} \pm 0.013$ | $0.450 \pm 0.006$ | - | $0.275 \pm 0.002$ | $0.350 \pm 0.067$ |
| mTANs | * | $0.287 \pm 0.005$ | $0.436 \pm 0.005$ | * | $0.268 \pm 0.018$ | $\mathbf{0.337} \pm 0.033$ |
| CSDI(n=5) | * | $0.287 \pm 0.003$ | $\mathbf{0.423} \pm 0.003$ | * | $0.378 \pm 0.001$ | $0.364 \pm 0.036$ |
| CSDI(n=25) | * | $\mathbf{0.270} \pm 0.001$ | $\mathbf{0.423} \pm 0.006$ | * | $0.340 \pm 0.000$ | $0.347 \pm 0.036$ |
| VAE | $1.427 \pm 0.001$ | $1.016 \pm 0.002$ | $0.524 \pm 0.006$ | $1.462 \pm 0.002$ | $1.086 \pm 0.004$ | $0.467 \pm 0.034$ |
| HI-VAE | $1.081 \pm 0.003$ | $0.321 \pm 0.008$ | $0.464 \pm 0.008$ | $1.078 \pm 0.005$ | $0.317 \pm 0.010$ | $0.380 \pm 0.060$ |
| GP-VAE | $1.077 \pm 0.006$ | $0.316 \pm 0.011$ | $0.463 \pm 0.008$ | $1.078 \pm 0.005$ | $0.318 \pm 0.010$ | $0.385 \pm 0.051$ |
| Ours(Uniform) | $1.063 \pm 0.001$ | $0.291 \pm 0.004$ | $0.445 \pm 0.003$ | $1.052 \pm 0.001$ | $\mathbf{0.265} \pm 0.002$ | $0.351 \pm 0.060$ |
| Ours(Periodic) | $\mathbf{1.060} \pm 0.002$ | $\mathbf{0.283} \pm 0.004$ | $0.443 \pm 0.004$ | $\mathbf{1.049} \pm 0.002$ | $\mathbf{0.260} \pm 0.003$ | $\mathbf{0.327} \pm 0.022$ |

(a) Beijing        (b) US Local        (c) Physionet2012

Figure 3: Comparison of the imputed values for examples in (a) Beijing, (b) US Local, and (c) Physionent2012 datasets, highlighting that TimeCIB provides more accurate imputations by considering temporal dependencies. Dots and crosses are observed and missing ground-truth values, respectively.

performance of our method is further enhanced when equipped with a temporal periodic kernel (10), highlighting our model's ability to incorporate the correct inductive bias.

**Imputation for electrical health records.** Time series imputation is of special importance in healthcare where each feature may have a distinct sampling period and strategies. In this context, we evaluate imputation methods on *Physionet2012 – Mortality Prediction Challenge* (Silva et al., 2012), which aims to predict in-hospital mortality of intensive care unit (ICU) patients from 48 hours of records with roughly $80\%$ of missing features. Furthermore, we conduct additional evaluations to assess whether the imputation methods preserve the critical characteristics of a given time series – i.e., whether a patient's status is

Table 3: Imputation and prediction performance on the clinical dataset.

| Methods | Physionet2012 (mortality prediction)) | | |
|---|---|---|---|
| | NLL($\downarrow$) | MSE($\downarrow$) | AUROC($\uparrow$) |
| No Imp. | - | $0.962 \pm 0.000$ | $0.692 \pm 0.000$ |
| Mean Imp. | - | $0.511 \pm 0.000$ | $0.703 \pm 0.000$ |
| Forward Imp. | - | $0.613 \pm 0.000$ | $0.710 \pm 0.000$ |
| BRITS | - | $0.529 \pm 0.004$ | $0.700 \pm 0.005$ |
| SAITS | - | $0.501 \pm 0.024$ | $0.713 \pm 0.007$ |
| mTANs | * | $\mathbf{0.499} \pm 0.008$ | $0.721 \pm 0.004$ |
| CSDI(n=5) | * | $0.548 \pm 0.014$ | $0.705 \pm 0.005$ |
| CSDI(n=25) | * | $\mathbf{0.478} \pm 0.002$ | $0.683 \pm 0.033$ |
| VAE | $1.400 \pm 0.000$ | $0.962 \pm 0.000$ | $0.691 \pm 0.001$ |
| HI-VAE | $1.345 \pm 0.009$ | $0.852 \pm 0.018$ | $0.696 \pm 0.004$ |
| GP-VAE | $1.227 \pm 0.007$ | $0.616 \pm 0.013$ | $0.730 \pm 0.006$ |
| Ours(Uniform) | $\mathbf{1.183} \pm 0.007$ | $0.528 \pm 0.014$ | $\mathbf{0.744} \pm 0.009$ |
| Ours(Cauchy) | $\mathbf{1.179} \pm 0.006$ | $0.521 \pm 0.012$ | $\mathbf{0.744} \pm 0.009$ |

deteriorating or not – after replacing the missing features with imputed values. Table 3 shows that TimeCIB provides imputation performance comparable to the best benchmark while outperforming the VAE-based methods by a great margin. Furthermore, it achieves the best classification performance, successfully capturing information about the temporal dynamics of patients' status. Note that while the mean imputation provides better imputation performance, the imputed values drastically lose the crucial information for discriminating patient's status.

## 6 CONCLUSION

In this paper, we have presented TimeCIB, a novel information-theoretic approach for time series imputation. While inheriting the multiple imputation and uncertainty measurement properties of the IB, TimeCIB addresses the limitation of the IB principle in capturing underlying temporal dynamics by replacing conventional regularization with conditional regularization. Our variational decomposition showed that CIB could be approximated by optimizing the evidence lower bound (ELBO) and the contrastive objective. We also demonstrated that introducing inductive bias based on temporal kernels can further enhance expressive power, acting as a form of conditional alignment. Our empirical results on image sequences, weather forecasting, and electrical health records prove that TimeCIB is effective in a wide range of practical cases.

ACKNOWLEDGMENTS

We thank anonymous reviewers for many insightful comments and suggestions. CL was supported through the IITP grant funded by the Korea government(MSIT) (No. 2021-0-01341, AI Graduate School Program, CAU).

CODE AVAILABILITY

Codebase used in this paper is available at `https://github.com/Chemgyu/TimeCIB`.

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

# A DERIVATIONS

## A.1 VARIATIONAL APPROXIMATION OF FIRST TERM IN CONDITIONAL RECONSTRUCTION.

Here, we show that the first term of the conditional reconstruction $I(\mathbf{X}_t; \mathbf{Z}_t, \mathbf{X}^o_{\setminus t})$ can be approximated as in (4) based on the following derivation:

$$
\begin{aligned}
I(\mathbf{X}_t; \mathbf{Z}_t, \mathbf{X}^o_{\setminus t}) &= \mathbb{E}_{p(\mathbf{x}_t, \mathbf{z}_t, \mathbf{x}^o_{\setminus t})} \left[ \log \frac{p(\mathbf{x}_t | \mathbf{z}_t, \mathbf{x}^o_{\setminus t})}{p(\mathbf{x}_t)} \right] \\
&= \mathbb{E}_{p(\mathbf{x}_t, \mathbf{z}_t, \mathbf{x}^o_{\setminus t})} \left[ \log p(\mathbf{x}_t | \mathbf{z}_t, \mathbf{x}^o_{\setminus t}) \right] - \mathbb{E}_{p(\mathbf{x}_t)} \left[ \log p(\mathbf{x}_t) \right] \\
&= \int_{\mathbf{x}_t} \int_{\mathbf{z}_t} \int_{\mathbf{x}^o_{\setminus t}} p(\mathbf{x}_t, \mathbf{z}_t, \mathbf{x}^o_{\setminus t}) \log p(\mathbf{x}_t | \mathbf{z}_t, \mathbf{x}^o_{\setminus t}) d\mathbf{x}_t d\mathbf{z}_t d\mathbf{x}^o_{\setminus t} + \mathcal{H}(\mathbf{x}_t) \\
&\geq \int_{\mathbf{x}_t} \int_{\mathbf{z}_t} \int_{\mathbf{x}^o_{\setminus t}} p(\mathbf{x}_t, \mathbf{z}_t, \mathbf{x}^o_{\setminus t}) \log p(\mathbf{x}_t | \mathbf{z}_t, \mathbf{x}^o_{\setminus t}) d\mathbf{x}_t d\mathbf{z}_t d\mathbf{x}^o_{\setminus t}
\end{aligned}
\tag{11}
$$

The last inequality holds because of the non-negativity of entropy.

Note that $p(\mathbf{X}_t, \mathbf{Z}_t, \mathbf{X}^o_{\setminus t})$ can be marginalized by introducing $\mathbf{X}^o_t$:

$$
p(\mathbf{X}_t, \mathbf{Z}_t, \mathbf{X}^o_{\setminus t}) = \int_{\mathbf{X}^o_t} p(\mathbf{X}_t, \mathbf{X}^o_{1:T}) q_\phi(\mathbf{Z}_t | \mathbf{X}^o_{1:T}) d\mathbf{X}^o_t
\tag{12}
$$

Then, putting (12) into the (11) gives

$$
\begin{aligned}
I(\mathbf{X}_t; \mathbf{Z}_t, \mathbf{X}^o_{\setminus t}) &\geq \int_{\mathbf{x}_t} \int_{\mathbf{z}_t} \int_{\mathbf{x}^o_{1:T}} p(\mathbf{x}_t, \mathbf{x}^o_{1:T}) q_\phi(\mathbf{z}_t | \mathbf{x}^o_{1:T}) \log p(\mathbf{x}_t | \mathbf{z}_t, \mathbf{x}^o_{\setminus t}) d\mathbf{x}_t d\mathbf{z}_t d\mathbf{x}^o_{1:T} \\
&= \mathbb{E}_{p(\mathbf{x}_t, \mathbf{x}^o_{1:T})} \left[ \mathbb{E}_{q_\phi(\mathbf{z}_t | \mathbf{x}^o_{1:T})} \left[ \log p(\mathbf{x}_t | \mathbf{z}_t, \mathbf{x}^o_{\setminus t}) \right] \right] \\
&= \mathbb{E}_{p(\mathbf{x}_t, \mathbf{x}^o_{1:T})} \left[ \mathbb{E}_{q_\phi(\mathbf{z}_t | \mathbf{x}^o_{1:T})} \left[ \log p(\mathbf{x}_t | \mathbf{z}_t, \mathbf{x}^o_{\setminus t}) \frac{p_\theta(\mathbf{x}_t | \mathbf{z}_t)}{p_\theta(\mathbf{x}_t | \mathbf{z}_t)} \right] \right] \\
&= \mathbb{E}_{p(\mathbf{x}_t, \mathbf{x}^o_{1:T})} \left[ \mathbb{E}_{q_\phi(\mathbf{z}_t | \mathbf{x}^o_{1:T})} \left[ \log p_\theta(\mathbf{x}_t | \mathbf{z}_t) \right] \right] + \mathbb{E}_{p(\mathbf{x}_t, \mathbf{z}_t, \mathbf{x}^o_{\setminus t})} \left[ \log \frac{p(\mathbf{x}_t | \mathbf{z}_t, \mathbf{x}^o_{\setminus t})}{p_\theta(\mathbf{x}_t | \mathbf{z}_t)} \right]
\end{aligned}
\tag{13}
$$

The last term can be expressed in a form of the KL-divergence:

$$
\begin{aligned}
\mathbb{E}_{p(\mathbf{x}_t, \mathbf{z}_t, \mathbf{x}^o_{\setminus t})} \left[ \log \frac{p(\mathbf{x}_t | \mathbf{z}_t, \mathbf{x}^o_{\setminus t})}{p_\theta(\mathbf{x}_t | \mathbf{z}_t)} \right] &= \mathbb{E}_{p(\mathbf{z}_t, \mathbf{x}^o_{\setminus t})} \left[ \mathbb{E}_{p(\mathbf{x}_t | \mathbf{z}_t, \mathbf{x}^o_{\setminus t})} \left[ \log \frac{p(\mathbf{x}_t | \mathbf{z}_t, \mathbf{x}^o_{\setminus t})}{p_\theta(\mathbf{x}_t | \mathbf{z}_t)} \right] \right] \\
&= \mathbb{E}_{p(\mathbf{z}_t, \mathbf{x}^o_{\setminus t})} \left[ D_{\mathrm{KL}}(p(\mathbf{x}_t | \mathbf{z}_t, \mathbf{x}^o_{\setminus t}) || p_\theta(\mathbf{x}_t | \mathbf{z}_t)) \right]
\end{aligned}
\tag{14}
$$

Due to the non-negativity of the KL-divergence, plugging (14) into (13) gives (4) as

$$
I(\mathbf{X}_t; \mathbf{Z}_t, \mathbf{X}^o_{\setminus t}) \geq \mathbb{E}_{p(\mathbf{X}_t, \mathbf{X}^o_{1:T})} \left[ \mathbb{E}_{q_\phi(\mathbf{Z}_t | \mathbf{X}^o_{1:T})} \left[ \log p_\theta(\mathbf{X}_t | \mathbf{Z}_t) \right] \right] .
\tag{15}
$$

## A.2 VARIATIONAL APPROXIMATION OF SECOND TERM IN CONDITIONAL RECONSTRUCTION.

Analogous to the derivation in (Alemi et al., 2017), $I(\mathbf{X}_t; \mathbf{X}^o_{\setminus t})$ is upper bounded as follows:

$$
\begin{aligned}
I(\mathbf{X}_t; \mathbf{X}^o_{\setminus t}) &= \mathbb{E}_{p(\mathbf{x}_t, \mathbf{x}^o_{\setminus t})} \left[ \log \frac{p(\mathbf{x}_t | \mathbf{x}^o_{\setminus t})}{p(\mathbf{x}_t)} \right] \\
&= \mathbb{E}_{p(\mathbf{x}_t, \mathbf{x}^o_{\setminus t})} \left[ \log \frac{p(\mathbf{x}_t | \mathbf{x}^o_{\setminus t}) q(\mathbf{x}_t)}{p(\mathbf{x}_t) q(\mathbf{x}_t)} \right] \\
&= \mathbb{E}_{p(\mathbf{x}_t, \mathbf{x}^o_{\setminus t})} \left[ \log \frac{p(\mathbf{x}_t | \mathbf{x}^o_{\setminus t})}{q(\mathbf{x}_t)} \right] - \mathbb{E}_{p(\mathbf{x}_t, \mathbf{x}^o_{\setminus t})} \left[ \log \frac{p(\mathbf{x}_t)}{q(\mathbf{x}_t)} \right] \\
&= \mathbb{E}_{p(\mathbf{x}^o_{\setminus t})} \left[ D_{\mathrm{KL}}(p(\mathbf{x}_t | \mathbf{x}^o_{\setminus t}) || q(\mathbf{x}_t)) \right] - D_{\mathrm{KL}}(p(\mathbf{x}_t) || q(\mathbf{x}_t)) \\
&\leq \mathbb{E}_{p(\mathbf{x}^o_{\setminus t})} \left[ D_{\mathrm{KL}}(p(\mathbf{x}_t | \mathbf{x}^o_{\setminus t}) || q(\mathbf{x}_t)) \right]
\end{aligned}
\tag{16}
$$

where we set $q(\mathbf{X}_t) = \mathcal{N}(0, I)$ (Alemi et al., 2017; Lee et al., 2023). Specifically, in our implementation, we deterministically encode $\tilde{\mathbf{Z}}_t$ by taking $\mu_\phi(\mathbf{X}^o_{\setminus t})$ and then calculate the KL-divergence between the decoder distribution $p_\theta(\mathbf{X}_t | \tilde{\mathbf{Z}}_t)$ with $q(\mathbf{X}_t)$.

## A.3 VARIATIONAL APPROXIMATION OF FIRST TERM IN CONDITIONAL REGULARIZATION.

$$
\begin{aligned}
I(\mathbf{Z}_t; \mathbf{X}^o_{1:T}) &= \mathbb{E}_{q_\phi(\mathbf{z}_t, \mathbf{x}^o_{1:T})} \left[ \log \frac{q_\phi(\mathbf{x}^o_{1:T}, \mathbf{z}_t)}{q_\phi(\mathbf{z}_t) p_{\mathrm{data}}(\mathbf{x}^o_{1:T})} \right] \\
&= \mathbb{E}_{q_\phi(\mathbf{z}_t, \mathbf{x}^o_{1:T})} \left[ \log \frac{q_\phi(\mathbf{z}_t | \mathbf{x}^o_{1:T})}{q_\phi(\mathbf{z}_t)} \frac{p(\mathbf{z}_t)}{p(\mathbf{z}_t)} \right] \\
&= \mathbb{E}_{q_\phi(\mathbf{z}_t, \mathbf{x}^o_{1:T})} \left[ \log \frac{q_\phi(\mathbf{z}_t | \mathbf{x}^o_{1:T})}{p(\mathbf{z}_t)} \right] + \mathbb{E}_{q_\phi(\mathbf{z}_t, \mathbf{x}^o_{1:T})} \left[ \log \frac{p(\mathbf{z}_t)}{q_\phi(\mathbf{z}_t)} \right] \\
&= \mathbb{E}_{p_{\mathrm{data}}(\mathbf{x}^o_{1:T})} \left[ D_{\mathrm{KL}}(q_\phi(\mathbf{z}_t | \mathbf{x}^o_{1:T}) || p(\mathbf{z}_t)) \right] - D_{\mathrm{KL}}(q_\phi(\mathbf{z}_t) || p(\mathbf{z}_t)) \\
&\leq \mathbb{E}_{p_{\mathrm{data}}(\mathbf{x}^o_{1:T})} \left[ D_{\mathrm{KL}}(q_\phi(\mathbf{z}_t | \mathbf{x}^o_{1:T}) || p(\mathbf{z}_t)) \right]
\end{aligned}
\tag{17}
$$

The last inequality holds because of the non-negativity of KL-divergence.

A.4 CONTRASTIVE APPROXIMATION OF SECOND TERM IN CONDITIONAL REGULARIZATION.

In Section 3.2.2, we optimize $I(\mathbf{Z}_t; \mathbf{X}_{\backslash t}^o)$ by approximate the mutual information into a contrastive form, which is similar to (Oord et al., 2018; Tian et al., 2020).

$$
\begin{aligned}
I(\mathbf{Z}_t; \mathbf{X}_{\backslash t}^o) &= -\mathbb{E}_X \log \left[ \frac{p(\mathbf{X}_{\backslash t}^o)}{p(\mathbf{X}_{\backslash t}^o | \mathbf{Z}_t)} \right] \\
&= -\mathbb{E}_X \log \left[ \frac{p(\mathbf{X}_{\backslash t}^o)}{p(\mathbf{X}_{\backslash t}^o | \mathbf{Z}_t)} N \right] + \log(N) \\
&\geq -\mathbb{E}_X \log \left[ \frac{p(\mathbf{X}_{\backslash t}^o)}{p(\mathbf{X}_{\backslash t}^o | \mathbf{Z}_t)} N \right] \\
&\geq -\mathbb{E}_X \log \left[ 1 + \frac{p(\mathbf{X}_{\backslash t}^o)}{p(\mathbf{X}_{\backslash t}^o | \mathbf{Z}_t)} (N-1) \right] \\
&= -\mathbb{E}_X \log \left[ 1 + \frac{p(\mathbf{X}_{\backslash t}^o)}{p(\mathbf{X}_{\backslash t}^o | \mathbf{Z}_t)} (N-1) \mathbb{E}_{\mathbf{X}_{\backslash t}^{o,j}} \frac{p(\mathbf{X}_{\backslash t}^{o,j} | \mathbf{Z}_t)}{\mathbf{X}_{\backslash t}^{o,j}} \right] \\
&\approx -\mathbb{E}_X \log \left[ 1 + \frac{p(\mathbf{X}_{\backslash t}^o)}{p(\mathbf{X}_{\backslash t}^o | \mathbf{Z}_t)} \sum_{\mathbf{X}_{\backslash t}^- \in \mathcal{X}^-} \frac{p(\mathbf{X}_{\backslash t}^- | \mathbf{Z}_t)}{p(\mathbf{X}_{\backslash t}^-)} \right] \\
&= \mathbb{E}_X \log \left[ \frac{\frac{p(\mathbf{X}_{\backslash t}^o | \mathbf{Z}_t)}{p(\mathbf{X}_{\backslash t}^o)}}{\frac{p(\mathbf{X}_{\backslash t}^o | \mathbf{Z}_t)}{p(\mathbf{X}_{\backslash t}^o)} + \sum_{\mathbf{X}_{\backslash t}^- \in \mathcal{X}^-} \frac{p(\mathbf{X}_{\backslash t}^- | \mathbf{Z}_t)}{p(\mathbf{X}_{\backslash t}^-)}} \right] \\
&\approx \mathbb{E}_X \log \left[ \frac{\frac{p(\mathbf{X}_{\backslash t}^o | \mathbf{Z}_t)}{p(\mathbf{X}_{\backslash t}^o)}}{\sum_{\mathbf{X}_{1:T}^- \in \mathcal{X}^-} \frac{p(\mathbf{X}_{1:T}^- | \mathbf{Z}_t)}{p(\mathbf{X}_{1:T}^-)}} \right] \\
&= \mathbb{E}_X \log \left[ \frac{f(\mathbf{Z}_t, \mathbf{X}_{\backslash t}^o)}{\sum_{\mathbf{X}_{1:T}^- \in \mathcal{X}^-} f(\mathbf{Z}_t, \mathbf{X}_{1:T}^-)} \right]
\end{aligned}
\tag{18}
$$

There remain two design choices: i) selection of $\mathcal{X}^-$ and ii) formulation of function $f$. For i), we use a mini-batch approach that $\mathbf{X}_{1:T}^-$ are chosen from other time series inputs in the same mini-batch. For ii), we adopt the average of cosine similarities along the time axis. Specifically, we define the function $f$ using an alternative representations $\tilde{\mathbf{Z}}_{t'}$ which is obtained by inputting the masked input $q_\phi(\tilde{\mathbf{Z}}_{t'} | \mathbf{X}_{\backslash t}^o)$:

$$
f(\mathbf{Z}_t, \mathbf{X}_{\backslash t}^o) = \sum_{t' \in \{1:T\} \backslash t} f(\mathbf{Z}_t, \tilde{\mathbf{Z}}_{t'}) = \sum_{t' \in \{1:T\} \backslash t} \exp \left( \mathbf{Z}_t^T \tilde{\mathbf{Z}}_{t'} / \tau \right)
\tag{19}
$$

where $\tau \in \Re$ is a temperature hyperparameter.

Alternatively, we can also define the function for negative samples in a similar way.

$$
f(\mathbf{Z}_t, \mathbf{X}_{1:T}^-) = \sum_{t' \in \{1:T\}} f(\mathbf{Z}_t, \mathbf{Z}_{t'}^-) = \sum_{t' \in \{1:T\}} \exp \left( \mathbf{Z}_t^T \mathbf{Z}_{t'}^- / \tau \right)
\tag{20}
$$

Then $I(\mathbf{Z}_t; \mathbf{X}_{\backslash t}^o)$ is lower bounded by

$$
I(\mathbf{Z}_t; \mathbf{X}_{\backslash t}^o) \geq \log \left[ \frac{\sum_{t' \in \{1:T\} \backslash t} \exp \left( \mathbf{Z}_t^T \tilde{\mathbf{Z}}_{t'} / \tau \right)}{\sum_{\mathbf{X}_{1:T}^- \in \mathcal{X}_{1:T}^-} \sum_{t' \in \{1:T\}} \exp \left( \mathbf{Z}_t^T \mathbf{Z}_{t'}^- / \tau \right)} \right]
\tag{21}
$$

We can observe that $I(\mathbf{Z}_t; \mathbf{X}_{\backslash t}^o)$ is lower bounded by the similar form of NT-Xent objective function in (Sohn, 2016) and specifically *in*-version of the supervised contrastive loss in (Khosla et al., 2020).

# B ADDITIONAL RESULTS

## B.1 MORE ON ROBUSTNESS

**Missing Patterns.** To assess the robustness of our model on diverse missing patterns, we tested our model on five different types of missingness generated by the missing mechanism outlined in (Fortuin et al., 2020). Here, we describe the data-generating process of each missing pattern as the folowing: *Spatial*: Define a spatial Gaussian process with RBF kernel, and draw the missingness patterns by sampling from this process. *Temporal+*: Define a temporal Gaussian process with RBF kernel for each feature dimension, and draw from this process. *Temporal-*: Implement with determinantial point process for each feature dimension.

Table B1: Robustness on missing patterns, HealingMNIST (NLL)

| Pattern | Mean imp. | Forward imp. | VAE | HI-VAE | GP-VAE | Ours |
|---|---|---|---|---|---|---|
| Random | - | - | $0.3360 \pm 0.0010$ | $0.2218 \pm 0.0008$ | $0.1957 \pm 0.0006$ | $\mathbf{0.0887} \pm 0.0010$ |
| Spatial | - | - | $0.4802 \pm 0.0016$ | $0.2259 \pm 0.0010$ | $0.1779 \pm 0.0007$ | $\mathbf{0.1509} \pm 0.0038$ |
| Temporal- | - | - | $0.1940 \pm 0.0006$ | $0.1349 \pm 0.0005$ | $0.1175 \pm 0.0004$ | $\mathbf{0.0879} \pm 0.0004$ |
| Temporal+ | - | - | $0.1918 \pm 0.0006$ | $0.1332 \pm 0.0005$ | $0.1206 \pm 0.0004$ | $\mathbf{0.0927} \pm 0.0006$ |
| MNAR | - | - | $0.4798 \pm 0.0016$ | $0.2896 \pm 0.0010$ | $0.2606 \pm 0.0008$ | $\mathbf{0.2017} \pm 0.0041$ |

Table B2: Robustness on missing patterns, HealingMNIST (MSE)

| Pattern | Mean imp. | Forward imp. | VAE | HI-VAE | GP-VAE | Ours |
|---|---|---|---|---|---|---|
| Random | $0.069 \pm 0.000$ | $0.099 \pm 0.000$ | $0.100 \pm 0.000$ | $0.046 \pm 0.001$ | $\mathbf{0.036} \pm 0.000$ | $\mathbf{0.036} \pm 0.000$ |
| Spatial | $0.090 \pm 0.000$ | $0.099 \pm 0.000$ | $0.122 \pm 0.000$ | $0.097 \pm 0.000$ | $0.071 \pm 0.001$ | $\mathbf{0.059} \pm 0.000$ |
| Temporal- | $0.086 \pm 0.000$ | $0.093 \pm 0.000$ | $0.101 \pm 0.000$ | $0.046 \pm 0.001$ | $0.037 \pm 0.001$ | $\mathbf{0.036} \pm 0.000$ |
| Temporal+ | $0.107 \pm 0.000$ | $0.117 \pm 0.000$ | $0.101 \pm 0.000$ | $0.047 \pm 0.001$ | $0.038 \pm 0.001$ | $\mathbf{0.037} \pm 0.000$ |
| MNAR | $0.168 \pm 0.000$ | $0.177 \pm 0.000$ | $0.232 \pm 0.000$ | $0.134 \pm 0.003$ | $0.114 \pm 0.002$ | $\mathbf{0.088} \pm 0.002$ |

**Missing Ratio.** Analogous to (Fortuin et al., 2020), we validate the robustness of our model on a variety of different missing ratios using random missing patterns, ranging from 10% to 90%.

Table B3: Robustness on random missingness, HealingMNIST (NLL)

| Missing | VAE | HI-VAE | GP-VAE | Ours |
|---|---|---|---|---|
| 10 % | $0.1319 \pm 0.0039$ | $0.0783 \pm 0.0022$ | $0.0695 \pm 0.0010$ | $\mathbf{0.0666} \pm 0.0004$ |
| 20 % | $0.1506 \pm 0.0054$ | $0.0844 \pm 0.0009$ | $0.0744 \pm 0.0009$ | $\mathbf{0.0688} \pm 0.0004$ |
| 30 % | $0.1637 \pm 0.0042$ | $0.0874 \pm 0.0057$ | $0.0799 \pm 0.0008$ | $\mathbf{0.0734} \pm 0.0005$ |
| 40 % | $0.1950 \pm 0.0061$ | $0.0974 \pm 0.0102$ | $0.0864 \pm 0.0007$ | $\mathbf{0.0782} \pm 0.0003$ |
| 50 % | $0.2164 \pm 0.0046$ | $0.1172 \pm 0.0127$ | $0.0937 \pm 0.0012$ | $\mathbf{0.0849} \pm 0.0005$ |
| 60 % | $0.2634 \pm 0.0108$ | $0.1264 \pm 0.0070$ | $0.1040 \pm 0.0008$ | $\mathbf{0.0944} \pm 0.0001$ |
| 70 % | $0.2800 \pm 0.0051$ | $0.1489 \pm 0.0034$ | $0.1197 \pm 0.0003$ | $\mathbf{0.1088} \pm 0.0002$ |
| 80 % | $0.2928 \pm 0.0058$ | $0.1698 \pm 0.0177$ | $0.1408 \pm 0.0002$ | $\mathbf{0.1317} \pm 0.0009$ |
| 90 % | $0.3071 \pm 0.0054$ | $0.2290 \pm 0.0042$ | $0.1852 \pm 0.0003$ | $\mathbf{0.1691} \pm 0.0006$ |

Table B4: Robustness on random missingness, HealingMNIST (MSE)

| Missing | VAE | HI-VAE | GP-VAE | Ours |
|---|---|---|---|---|
| 10 % | $0.0582 \pm 0.0023$ | $0.0336 \pm 0.0009$ | $0.0295 \pm 0.0004$ | $\mathbf{0.0274} \pm 0.0008$ |
| 20 % | $0.0700 \pm 0.0030$ | $0.0367 \pm 0.0000$ | $0.0311 \pm 0.0004$ | $\mathbf{0.0283} \pm 0.0001$ |
| 30 % | $0.0857 \pm 0.0033$ | $0.0373 \pm 0.0029$ | $0.0336 \pm 0.0007$ | $\mathbf{0.0305} \pm 0.0002$ |
| 40 % | $0.1185 \pm 0.0033$ | $0.0420 \pm 0.0051$ | $0.0362 \pm 0.0001$ | $\mathbf{0.0328} \pm 0.0001$ |
| 50 % | $0.1317 \pm 0.0006$ | $0.0517 \pm 0.0063$ | $0.0390 \pm 0.0005$ | $\mathbf{0.0356} \pm 0.0001$ |
| 60 % | $0.1328 \pm 0.0000$ | $0.0561 \pm 0.0036$ | $0.0436 \pm 0.0001$ | $\mathbf{0.0402} \pm 0.0000$ |
| 70 % | $0.1328 \pm 0.0000$ | $0.0681 \pm 0.0018$ | $0.0505 \pm 0.0001$ | $\mathbf{0.0470} \pm 0.0000$ |
| 80 % | $0.1328 \pm 0.0000$ | $0.0787 \pm 0.0108$ | $0.0594 \pm 0.0001$ | $\mathbf{0.0584} \pm 0.0008$ |
| 90 % | $0.1328 \pm 0.0000$ | $0.1088 \pm 0.0076$ | $0.0826 \pm 0.0009$ | $\mathbf{0.0755} \pm 0.0002$ |

## B.2 ABLATION STUDY ON THE CONDITIONAL RECONSTRUCTION.

As derived in Section 3.2, the conditional reconstruction term is given as the follows: $I(\mathbf{X}_t; \mathbf{Z}_t | \mathbf{X}_{\setminus t}^o) = I(\mathbf{X}_t; \mathbf{Z}_t, \mathbf{X}_{\setminus t}^o) - I(\mathbf{X}_t; \mathbf{X}_{\setminus t}^o)$. To support our claim that the effect of minimizing $I(\mathbf{X}_t; \mathbf{X}_{\setminus t}^o)$ deteriorates the model performance, we introduce an auxiliary weight $\omega$ and make a weighted version of the conditional reconstruction term as

$$\tilde{I}(\mathbf{X}_t; \mathbf{Z}_t | \mathbf{X}_{\setminus t}^o) = I(\mathbf{X}_t; \mathbf{Z}_t, \mathbf{X}_{\setminus t}^o) - \omega I(\mathbf{X}_t; \mathbf{X}_{\setminus t}^o). \tag{22}$$

Then, we can see the effect of $I(\mathbf{X}_t; \mathbf{X}_{\setminus t}^o)$ by varying the value of $\omega$ from 0 to 1 by assessing the imputation performance on the Physionet2012 dataset. Figure B2 shows that, starting from $\omega = 0$ which is equivalent to the CIB objective in (8), focusing more on $I(\mathbf{X}_t; \mathbf{X}_{\setminus t}^o)$ gradually deteriorates both the imputation performance (MSE) and the classification performance (AUROC).

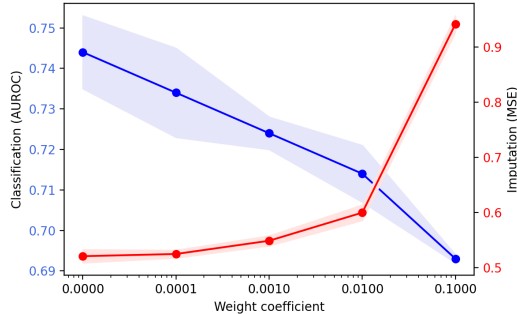

Figure B1: The effect of $\omega$ in the weighted conditional reconstruction in (22).

## B.3 QUANTITATIVE ASSESSMENT ON $I(\mathbf{Z}_t; \mathbf{X}_{\setminus t}^o)$

In addition to the qualitative assessment on $I(\mathbf{Z}_t; \mathbf{X}_{\setminus t}^o)$ shown in Figure 1 and empirical performances in Section 5, we compare the approximated quantity on $I(\mathbf{Z}_t; \mathbf{X}_{\setminus t}^o)$ using (9) to assert that our CIB objective indeed preserves more information. Please note that, as derived in (A.4), the lower bound is given by $I(\mathbf{Z}_t; \mathbf{X}_{\setminus t}^o) \geq \log N - \mathcal{L}_\phi^3$, which depends on the batch size $N$ and the objective $\mathcal{L}_\phi^3$. Table B5 shows the values of $\mathcal{L}_\phi^3$, indicating that a lower value corresponds to more preserved information. This result demonstrates that our CIB objective preserves more information in $I(\mathbf{Z}_t; \mathbf{X}_{\setminus t}^o)$ based on the approximation.

Table B5: Quantitative results on $I(\mathbf{Z}_t; \mathbf{X}_{\setminus t}^o)$. Lower value indicates more preservation of information.

| Dataset | VAE | HI-VAE | GP-VAE | Ours(Uniform) | Ours(Kernel) |
|---|---|---|---|---|---|
| RotatedMNIST | $3.619 \pm 0.005$ | $3.577 \pm 0.008$ | $3.435 \pm 0.001$ | $\mathbf{3.428} \pm 0.006$ | $\mathbf{3.418} \pm 0.002$ |
| Beijing | $3.950 \pm 0.015$ | $3.822 \pm 0.012$ | $3.762 \pm 0.012$ | $\mathbf{3.700} \pm 0.008$ | $\mathbf{3.686} \pm 0.006$ |

## B.4 SENSITIVITY ANALYSIS

Here, we provide the sensitivity analysis on $\beta$ and $\gamma$ in (8). Before presenting results, we rewrite our objective as follows to analyze the effects of individual regularization terms given fixed reconstruction,

$$\min_{\phi,\theta} \quad \mathcal{L}^1_{\phi,\theta} + \beta'\mathcal{L}^2_\phi + \gamma'\mathcal{L}^3_\phi \tag{23}$$

where $\beta' = 1/\beta$ and $\gamma' = \gamma/\beta$. Now, (23) shares the same notational convention used in $\beta$-VAE (Higgins et al., 2017), where we can systematically assess the sensitivity of TimeCIB on the two regularization objectives.

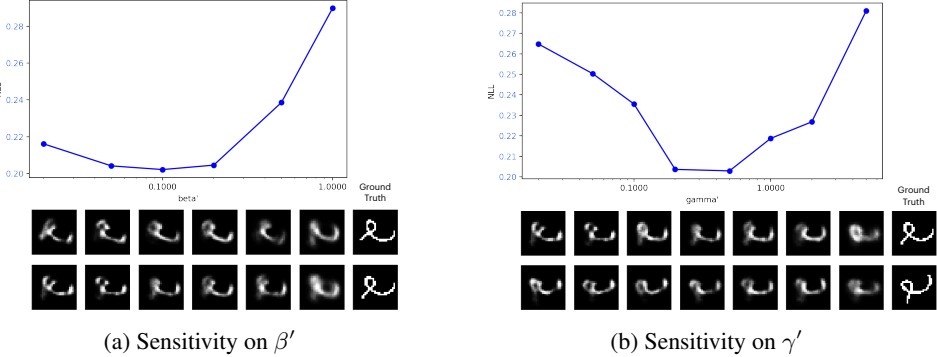

(a) Sensitivity on $\beta'$      (b) Sensitivity on $\gamma'$

Figure B2: Sensitivity analysis on the regularization objectives, $\beta'$ and $\gamma'$. $\gamma' = 0.5$ for (a), and $\beta' = 0.1$ for (b).

**Sensitivity on $\beta'$.** $\beta'$ is related to the regularization trade-off that low-$\beta'$ (weak regularization) can cause lack of timeseries-wise information and eventually overfitting on observed values, while high-$\beta'$ (strong regularization) can cause loss of timepoint-wise information since all distribution would become a unit Gaussian, as well-studied by previous works.

**Sensitivity on $\gamma'$.** Since $\gamma'$ also works as a regularization term, sensitivity on $\gamma'$ is similar to $\beta'$, but the cause of behaviour is slightly different. For low-$\gamma'$ (weak regularization), there is no driving force on conserving temporal dynamics (see Section 3.1), thereby cause loss of temporal dynamics and would eventually become HI-VAE (Nazabal et al., 2020) when $\gamma' = 0$. On the other hand, high-$\gamma'$ (strong regularization) cause loss of timepoint-wise information, since $L^3_\phi$ will map all $\mathbf{z}_t$ from the same time series into a single point, which means that we cannot distinguish individual timepoints. Figure B2 illustrates quantitative and qualitative results on the HealingMNIST.

**Optimization Strategy.** Simultaneous optimization of $\beta'$ and $\gamma'$ could be difficult. However, motivated from the above sensitivity analysis, one can first set $\gamma'=0$ and find optimal $\beta'$ (which is HI-VAE). Second-stage parameter search within the close range of order of magnitude of $\beta'$ could help finding optimal $\gamma'$.

## C  IMPLEMENTATION DETAILS

Here, we provide implementation details, including illustration of overall frameworks, algorithms, and hyperparameters. The overall framework is illustrated in Figure C1.

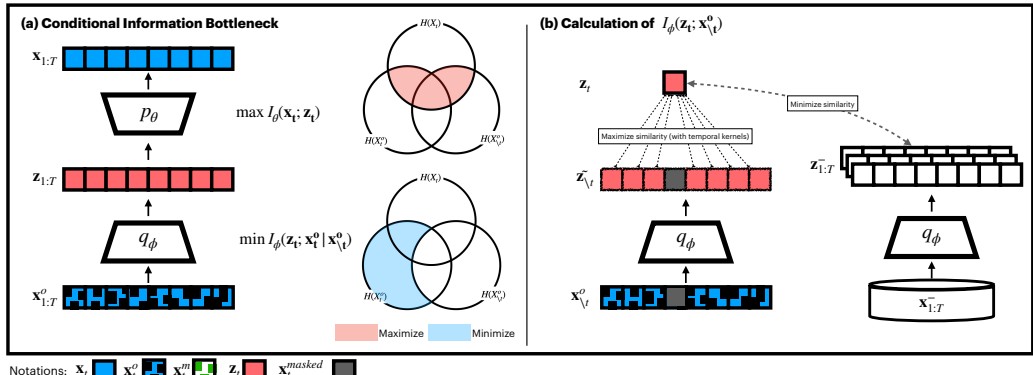

Figure C1: Illustration of TimeCIB framework.(A) Conceptual illustration TimeCIB framework. CIB maximizes $I_\theta(\mathbf{x}_t; \mathbf{z}_t)$, while minimizing $I_\phi(\mathbf{z}_t; \mathbf{x}_t^o|\mathbf{x}_{\backslash t}^o)$. (B) Implementation of optimizing $I_\phi(\mathbf{z}_t; \mathbf{x}_{\backslash t}^o)$.

### C.1  TRAINING AND INFERENCE.

At training stage, input time sequences are first fed into the stochastic encoder $q_\phi(\mathbf{Z}_t|\mathbf{X}_{1:T}^o) = \mathcal{N}(\mu_\phi(\mathbf{X}_{1:T}^o), \text{diag}(\sigma_\phi(\mathbf{X}_{1:T}^o)))$ where we use a 2 layers of bidirectional LSTM. For image dataset, we used CNN preprofessor before the encoder module. We then sample $\mathbf{z}_t$ from resulting latent distribution, and decoder $p_\theta(\mathbf{X}_t|\mathbf{Z}_t) = \mathcal{N}(\mu_\theta(\mathbf{z}_t), \sigma_\theta I)$ reconstructs the complete observation from the latent vector. In addition to previous conventional ELBO procedures, we also encodes $\mathbf{X}_{\backslash t}^o$, which are generated by masking $\mathbf{X}_t^o$ with zeros. These masked input series are also mapped by the same encoder and latent vectors are obtained. Consequently, we use two sets of time series representations: $\mathbf{z}_{1:T}$ and $\tilde{\mathbf{z}}_{1:T}$ for calculating (7). For training efficiency, we optimized (7) stochastically, which is, we randomly sampled $t$ at every epoch and minimized contrastive loss on that time step. At single prediction, we use the mean of the latent distribution $\mu_\phi(\mathbf{X}_{1:T}^o)$ as the corresponding latent representation and also reconstruct the observation from the mean of the decoder distribution $\mu_\theta(\mathbf{z}_t)$. All experiments were conducted using a 48GB NVIDIA RTX A6000.

## C.2 ALGORITHM

We provide the pseudo-algorithm of TimeCIB as below:

---

**Algorithm 1** Conditional Information Bottleneck on Time Series

---

**input** batch size $B$, time length $T$
**for** sampled minibatch $\{\mathbf{X}_{1:T}^o\}_1^B$ **do**
    **sample** $t \in \{1:T\}$
    generate masked input $\{\mathbf{X}_{\setminus t}^o\}_1^B$
    **for** $k \in \{1, \ldots, B\}$ **in parallel do**
        **for** $t' \in \{1:T\}$ **in parallel do**
            **encode** and **sample** $\mathbf{Z}_{t'}^k \sim q_\phi(\mathbf{Z}_{t'}^k | \mathbf{X}_{1:T}^{o,k}) = \mathcal{N}(\mu_\phi(\mathbf{X}_{1:T}^{o,k})\sigma_\phi(\mathbf{X}_{1:T}^{o,k})I)$
            **encode** and **sample** $\tilde{\mathbf{Z}}_{t'}^k \sim q_\phi(\tilde{\mathbf{Z}}_{t'}^k | \mathbf{X}_{\setminus t'}^{o,k}) = \mathcal{N}(\mu_\phi(\mathbf{X}_{\setminus t'}^{o,k}), \sigma_\phi(\mathbf{X}_{\setminus t'}^{o,k})I)$
            **decode** $p_\theta(\mathbf{X}_{t'}^k | \mathbf{Z}_{t'}) = \mathcal{N}(\mu_\theta(\mathbf{Z}_{t'}), \text{diag}(\sigma_\theta(\mathbf{Z}_{t'}))$
            $\mathcal{L}_{\phi,\theta}^1 = \mathcal{L}_{\phi,\theta}^1 - \log(p_\theta(\mathbf{X}_{t'}^{k,o} | \mathbf{Z}_{t'}))$
            $\mathcal{L}_\phi^2 = \mathcal{L}_\phi^2 + \text{KL}[q_\phi(\tilde{\mathbf{Z}}_{t'}^k | \mathbf{X}_{\setminus t'}^{o,k}) || \mathcal{N}(\mathbf{0}, I)]$
        **end for**
    **end for**
    **for** $k \in \{1, \ldots, B\}$**in parallel do**
        $\mathcal{L}_\phi^3 = \mathcal{L}_\phi^3 + \text{Contrastive Loss}(\mathbf{Z}_{t'}^k, \tilde{\mathbf{Z}}_{\setminus t}^k, \mathbf{Z}_{1:T}^{\setminus k})(9)$
    **end for**
    $\mathcal{L} = \beta\mathcal{L}_{\phi,\theta}^1 + \mathcal{L}_\phi^2 + \gamma\mathcal{L}_\phi^3$
    **update** $\phi, \theta$ to minimize $\mathcal{L}$
**end for**

---

## C.3 HYPERPARAMETER SPECIFICATION.

We provide hyperparameter settings used for our experiments.

Table C1: Hyperparameter specifications.

| | Hidden Dim | Batch Size | Epochs | Learning Rate | Temperature | Kernel parameter |
|---|---|---|---|---|---|---|
| HealingMNIST | 128 | 64 | 30 | $1e-3$ | 1.0 | 2.0 |
| RotatedMNIST | 128 | 64 | 30 | $1e-3$ | 1.0 | 2.0 |
| Beijing | 128 | 64 | 100 | $1e-3$ | 1.0 | 4.0 |
| US Local | 64 | 16 | 20 | $1e-4$ | 1.0 | 2.0 |
| Physionet2012 | 16 | 256 | 50 | $1e-3$ | 1.0 | 32 |

For a fair comparison, the magnitudes of the number of parameters are set to be equivalent among the evaluated deep learning methods.

Table C2: Number of learnable parameters.

| | Ours | VAE | HI-VAE | GP-VAE | BRITS | SAITS | mTANs | CSDI |
|---|---|---|---|---|---|---|---|---|
| HealingMNIST | $1.2M$ | $1.2M$ | $1.2M$ | $1.2M$ | - | - | - | - |
| RotatedMNIST | $1.2M$ | $1.2M$ | $1.2M$ | $1.2M$ | - | - | - | - |
| Beijing | $368K$ | $368K$ | $368K$ | $368K$ | $343K$ | $362K$ | $398K$ | $375K$ |
| US Local | $68K$ | $68K$ | $68K$ | $68K$ | $51K$ | $87K$ | $63K$ | $62K$ |
| Physionet2012 | $21K$ | $21K$ | $21K$ | $21K$ | $24K$ | $27K$ | $29K$ | $33K$ |

# D  DATASET, BASELINES, AND MORE RELATED WORKS

## D.1  DATASET STATISTICS

We provide baseline statistics of the benchmark datasets.

Table D1: Data Statistics.

|  | # Samples | Len ($T$) | Feature Dim | # Classes | Missing Ratio (ori/art) |
|---|---|---|---|---|---|
| HealingMNIST | 50000/10000/10000 | 10 | $28 \times 28 \times 1$ | 10 | - /60% |
| RotatedMNIST | 50000/10000/10000 | 10 | $28 \times 28 \times 1$ | 10 | - /60% |
| Beijing | 851/304/306 | 24 | 132 | - | 2%/60% |
| USLocal | 847/298/298 | 168 | 11 | - | -/60% |
| Physionet2012 | 3997/3993/3997 | 48 | 35 | 2 | 80%/ 2% |

## D.2  BASELINE MODELS

**HI-VAE (Nazabal et al., 2020)**  HI-VAE proposes a general generative framework that extends the VAE structure to be suitable for fitting incomplete heterogeneous data. Nazabal et al. (2020) proposed likelihood models for diverse data – real-valued, positive real-valued, interval, categorical, ordinal, and count data – and also expanded the conventional VAE to handle missing features. In this paper, we followed their approach to calculate reconstruction only to the observed features.

**GP-VAE (Fortuin et al., 2020)**  GP-VAE expands information bottleneck on time series imputation, by using Gaussian process prior in the latent space. To use the Gaussian process, GP-VAE's inference model encodes the input with respect to each latent dimension, $q(\mathbf{z}_{1:T,j}|\mathbf{x}_{1:T}^o) = \mathcal{N}(m_j, \Lambda_j^{-1})$, where $j$ indexes the dimensions in the latent space. The latent prior is formalized by $\mathbf{z}_t \sim \mathcal{GP}(m_z(\cdot), k_z(\cdot, \cdot))$ where $m$ and $k$ are the mean and covariance functions, respectively. GP-VAE also utilizes the Cauchy kernel as a covariance function to capture temporal dynamics.

## D.3  MORE RELATED WORKS

GAN-based imputation methods generate imputed values adversarially, assisted by a masked reconstruction loss, with the goal of making the generated samples closely resemble the original ones based on their observed values and indistinguishable from the original incomplete data by a discriminator. GAIN (Yoon et al., 2018a) uses a random masking and hinting mechanism to aid the discriminator in better distinguishing real and generated samples, thereby improving imputation performance. GRUI-GAN (Luo et al., 2018) and its extension $E^2$GAN (Luo et al., 2019) adopted the GAN framework into a temporal setting by enhancing the GRU cell to account for temporal dynamics, considering time lag influences related to observations and missing values.

# E  TABLE OF NOTATIONS

**Notations**

| | |
|---|---|
| $\mathbf{X}_{1:T}$ | Complete data features of one time series |
| $\mathbf{X}_{1:T}^{o}$ | Observed features of one time series |
| $\mathbf{X}_{1:T}^{m}$ | Missing features of one time series |
| $\mathbf{X}_{\setminus t}$ | Complete data features of one time series, except the time $t$ |
| $\mathbf{X}_{\setminus t}^{o}$ | Observed features of one time series, except the time $t$ |
| $\mathbf{X}_{\setminus t}^{m}$ | Missing features of one time series, exept the time $t$ |
| $\mathbf{X}_{t}$ | Complete data features at time $t$ |
| $\mathbf{X}_{t}^{o}$ | Observed features at time $t$ |
| $\mathbf{X}_{t}^{m}$ | Missing features at time $t$ |
| $\mathbf{Z}_{t}$ | Representation features at time $t$, generated from $\mathbf{X}_{1:T}^{o}$ |
| $\tilde{\mathbf{Z}}_{t}$ | Representation features at time $t$, generated from $\mathbf{X}_{\setminus t}^{o}$ |
| $\phi$ | Set of parameters of an encoder |
| $\theta$ | Set of parameters of a decoder |

