# OpenReview forum: "Conditional Information Bottleneck Approach for Time Series Imputation"
_ICLR.cc/2024/Conference — ICLR 2024 poster_

### Official Review · Reviewer_wRxf · 2023-10-18

**Soundness:** 4 excellent
**Presentation:** 3 good
**Contribution:** 4 excellent
**Rating:** 8
**Confidence:** 4

**Summary:**

This paper propose a novel time series imputation method inspired from the perspective of information bottleneck (IB). To solve the temporal context loss issue of vanilla IB, they propose a conditional IB to better consider temporal context. Several lower bound and upper bound are deducted for optimization. In a specific contrast learning loss, a kernel trick is used to reweright contribution of data from different time steps. Experiments results show that the proposed method achieve SOTA performance.

**Strengths:**

1. The method is inspired from a theoritical background of information bottleneck and can impute stochastically.
2. The bound derivations are interesting.

**Weaknesses:**

1. In equation 4, both the entropy term and the KL term are dropped to compute the lower bound. Will this bound be too loose? After dropping, the lower bound looks like a reconstruction loss rather than mutual information.

2. The paper doesn't mention the number of learnable parameters for each methods.

3. I'd like to see comparison with another two baselines mTAN [1] and CSDI [2].

[1] Multi-Time Attention Networks for Irregularly Sampled Time Series
[2] CSDI: Conditional Score-based Diffusion Models for Probabilistic Time Series Imputation

**Questions:**

None

---

> ### Author Response · Authors · 2023-11-16
> **Response to Reviewer wRxf**
>
> We thank the reviewer for the thoughtful comments and appreciation on our work. We have addressed all comments in the updated manuscript. Below, we provide a point-by-point response.
>
> ### **[W1] Loose bound on reconstruction**
>
> Please allow us to briefly summarize the discussion in [1] about the boundeness of (4). The first term is the entropy of the complete features at time step $t$. This quantity is independent from the parameters of our interest, so does not have any effect on the boundness. The second term is the KL-divergence between two distribution, $p(x_t|z_t)$ and $p_\theta(x_t|z_t)$, where $p_\theta(x_t|z_t)$ is a variational approximation of $p(x_t|z_t)$. The KL-divergence term will be small, resulting in a tight bound, if the assumption on the prior distribution $p(x_t|z_t)$ (in our case, a Gaussian distribution) is reasonable and the model achieves good reconstruction performance. For the detailed derivation and interpretation of (4), please refer to [1].
>
> ### **[W2] Hyperparameter details**
>
> We have provided a more detailed specification of hyperparameters, including the number of learnable parameters, in Appendix C of the revised version (page 20). To ensure a fair comparison, we ensure that the magnitudes of the number of parameters were set to be equivalent among the evaluated deep learning methods throughout our experiments.
>
>
> ### **[W3] Additional cutting-edge benchmarks**
>
> We have included the following **four** cutting-edge deep learning-based time series imputation  methods: two predictive methods including an RNN-based method, BRITS [1], and a transformer-based methods, SAITS [2], and two generative methods including an autoencoder-based method, mTANs [3], and a diffusion-based method, CSDI [4].
>
> Due to the limited rebuttal period, we have provided results for the Beijing, USLocal, and Physionet2012 datasets in the revised manuscript. Please find the result table on the general response and the revised manuscript. To summarize, our method achieves the best performance on the USLocal dataset, which has the longest time steps (T=168) and clear periodicity. For the Beijing dataset, our method performs slightly better than or comparably to state-of-the-art predictive methods. Here, CSDI (with n_samples=25) demonstrates the best performance but with a computation time approximately 2000 times slower than ours due to the required sampling process. For Physionet dataset, our method outperforms the cutting-edge benchmarks in terms of AUROC suggesting that our approach recovers semantically meaningful information for downstream tasks.
>
> ### References
>
> [1] Slava Voloshynovskiy, Mouad Kondah, Shideh Rezaeifar, Olga Taran, Taras Holotyak, and Danilo Jimenez Rezende. Information Bottleneck Through Variational Glasses. In *NeurIPS Workshop on Bayesian Deep Learning*, 2019.
>
> [2] Wei Cao, Dong Wang, Jian Li, Hao Zhou, Lei Li, and Yitan Li. BRITS: Bidirectional Recurrent Imputation for Time Series. *Advances in neural information processing systems*, 31, 2018.
>
> [3] Wenjie Du, David Cot́e, and Yan Liu. Saits: Self-attention-based imputation for time series. *Expert Systems with Applications*, 219:119619, 2023.
>
> [4] Satya Narayan Shukla and Benjamin Marlin. Multi-time attention networks for irregularly sampled time series. In *International Conference on Learning Representations*, 2021.
>
> [5] Yusuke Tashiro, Jiaming Song, Yang Song, and Stefano Ermon. CSDI: Conditional score-based diffusion models for probabilistic time series imputation. *Advances in Neural Information Processing Systems*, 34:24804–24816, 2021

---

> ### Author Response · Authors · 2023-11-20
> **Dear Reviewer wRxf**
>
> Once again, we would like to thank you for your invaluable feedback! We also appreciate the comments on other reviews above. We were wondering whether our response from November 16 has sufficiently addressed your questions. If you have any leftover comments or questions, please let us know - we would be happy to do our utmost to address them!
>
> Paper 4032 Authors

---

### Official Review · Reviewer_yyLt · 2023-10-30

**Soundness:** 3 good
**Presentation:** 3 good
**Contribution:** 3 good
**Rating:** 6
**Confidence:** 4

**Summary:**

Time series with missing values are prevalent in real-world applications. The paper proposes the use of an improved information bottleneck approach for irregular time series imputation, achieving superior performance compared to other VAE-based methods, as demonstrated in the experiments.

**Strengths:**

The model effectively addresses the issue of temporal information loss in the latent space of VAEs.

**Weaknesses:**

1. The paper exclusively focuses on Variational Autoencoder (VAE) models grounded in Information Bottleneck (IB) theory.  While this approach is well-articulated, it is notable that other kinds of generative models, for instance, ODE-based models and diffusion models, have demonstrated remarkable performance in the field of time series imputation.  The absence of a discussion or comparison with SOTA models in the related work and experimental sections is conspicuous, making it less convincing regarding its contribution to the irregular time series imputation task.

2. While the paper introduces a novel regularization method in the context of time series imputation, the innovation appears to be somewhat incremental. The approach can be perceived as a clever technique or ‘Trick’ rather than a substantial paradigm shift.

**Questions:**

1. What are the unique aspects of the CIB approach compared to other methodologies, including ODE, transformers, and diffusion models? Can you provide a more comprehensive evaluation, including comparisons with state-of-the-art models like ODE-based models and diffusion models, to demonstrate the performance of your model?

2. The paper claims that the CIB approach can mitigate the excessive regularization associated with IB. However, there seems to be a lack of quantitative analysis to substantiate this claim.  It would be beneficial if the authors could provide explicit metrics or criteria to measure the extent of regularization and demonstrate how CIB effectively addresses this issue.

3. Mitigating regularization is a double-edged sword, as it might lead to overfitting, especially in the context of learning temporal dependencies in time series data.  The paper should address how the CIB approach ensures an optimal balance, preventing the model from overly adapting to the training data and subsequently degrading its performance on unseen data.  Is there any empirical evidence or theoretical analysis to showcase this balance?

4. Using MINIST in the illustration of temporal dependency (Figure 1) is dubious compared to other irregular time series datasets (e.g., weather, traffic). It cannot demonstrate how the model handles temporal dependencies between different dimensions in multi-variate situations. How does the CIB approach model handle the multi-variate dependencies exactly?

---

> ### Author Response · Authors · 2023-11-16
> **Response to Reviewer yyLt (part 1/2).**
>
> We appreciate the reviewer for the insightful comments and suggestions. We have updated our paper in accordance with the reviewer’s feedback, addressing comments and questions as outlined below.
>
> ### **[Q1, W1] Comparison with other methods**
>
> Please allow us to first compare our method with predictive methods (including transformers) and diffusion methods, respectively. Comparing with predictive methods, our method supports multiple imputations and uncertainty quantification. This is extremely important in real-world data, because the confidence on individual imputed datapoint helps real-world decision-making, such as prescription for patients or weather forecasting. Comparing with diffusion methods, our method is much faster and allows us to introduce inductive bias about the underlying temporal dynamics, which is essential for real-time decision-makings and long-term periodic imputation. In our experiment, our method is approximately 2000 times faster than CSDI (with nsamples=25) under the same order of magnitude for the number of parameters. Our method could be more useful on the field that requires real-time imputation, such as weather forecasting or healthcare. Moreover, as shown in our experiments, we can inject inductive bias in terms of temporal kernels. This could provide more effective way to impute extremely long time series which has periodicity such as weather data, and our experiments on the USLocal dataset also supports this.
>
> We have included the following **four** cutting-edge deep learning-based time series imputation  methods: two predictive methods including an RNN-based method, BRITS [1], and a transformer-based methods, SAITS [2], and two generative methods including an autoencoder-based method, mTANs [3], and a diffusion-based method, CSDI [4].
>
> Due to the limited rebuttal period, we have provided results for the Beijing, USLocal, and Physionet2012 datasets in the revised manuscript. Please find the result table on the general response and the revised manuscript. To summarize, our method achieves the best performance on the USLocal dataset, which has the longest time steps (T=168) and clear periodicity. For the Beijing dataset, our method performs slightly better than or comparably to state-of-the-art predictive methods. Here, CSDI (with n_samples=25) demonstrates the best performance but with a computation time approximately 2000 times slower than ours due to the required sampling process. For Physionet dataset, our method outperforms the cutting-edge benchmarks in terms of AUROC suggesting that our approach recovers semantically meaningful information for downstream tasks.
>
> ### **[Q2,3] Discussions on Regularization**
>
> Before answer to these, we would like to clarify the difference between IB and CIB, in terms of how the CIB can alleviate regularization effect. It is true that CIB optimizes smaller amount of information, but it does not mean that the CIB imposes loose regularization. Rather, the CIB **guides** the direction of optimization, towards to preserve temporal dynamics, while having concise information.
>
> As illustrated in Figure 1(A) and (5), the absolute quantity of information that CIB optimizes is smaller than what the traditional IB does, i.e., ($I(Z_t; X^o_{1:T}) = I(Z_t; X^o_{1:T}) - I(Z_t;X^o_{\setminus t}$), $I(\cdot;\cdot) ≥ 0$). However, the more crucial aspect is the direction in which regularization guides the optimization, rather than the value itself. Since the traditional IB focuses on having concise information as illustrated in Figure 1(A), the direction of regularization does not consider preserving information about temporal dynamics from other time steps. Contrarily, the regularization using CIB effectively explicitly maintains information from the remaining time steps as derived in (5).
>
> We can provide the measured information $I(Z_t;X^o_{\setminus t}$) to demonstrate the conservation of temporal dynamics information. However, that metric may not offer useful insights as it is the objective we optimize. Instead, to showcase how well the temporal information is preserved and affects the downstream task, we use results on the RotatedMNIST dataset, where arbitrary timestep is totally missing such that the model must infer from the remaining time series. Therefore, we believe performance improvement on interpolation and extrapolation provides a proxy metric / criteria on how CIB effectively addresses this issue.

---

> ### Author Response · Authors · 2023-11-16
> **Response to Reviewer yyLt (part 2/2).**
>
> ### **[Q3] Generalizability**
>
> For generalizability, CIB provides more robust way to consider temporal dynamics. GP-VAE, which uses Gaussian Process under IB framework, lacks generalizability since it requires prior knowledge on the temporal smoothness (hyperparameters of Gaussian processes) and optimizes remaining parameters depending on that setting. Therefore, GP-VAE is vulnerable to input with totally different temporal dynamics. However, uniform CIB which uses no temporal kernels, provides effective way to impute time series without assumptions on the temporal dynamics of inputs (please refer to performances of (uniform) versions).
>
> ### **[Q4] Explanation on the MNIST experiments**
>
> **Details of Image Sequence Dataset.** We apologize for the lack of explanation of the dataset. The dataset we used - HealingMNIST, and RotatedMNIST - are ‘series of MNIST images’ (not a single MNIST) devised to reflect the real medical/clinical time series [5]. Specifically, every frame represents the collection of measurements that describe a current health of the patient, and the rotation of the images represents the progression of patient’s latent state. Since the latent logic of rotation is hidden in the input (only can observe the pixels), the model should be able to capture the temporal dynamics of rotating digits. These ‘series of images’ dataset have a strong advantage that we can visualize the results with diverse patterns/ratios of missingness (Figure 2). Moreover, these datasets are especially useful for qualitative analysis, please refer to the added sensitivity analysis on HealingMNIST, shown in Appendix B3 (page 18).
>
> **Multi-variate dependencies in CIB.** Our model first encodes the observed features (regarding multivariate dependencies) and map latent features, which represents the latent state of the sample.  On the latent representation space, the representations of individual timestep models smooth or periodic progression of latent states by taking advantage of conditional information objective. From these learned representations possessing enhanced temporal dynamics information, the decoder reconstructs the complete input.
>
> ### References
>
> [1] Wei Cao, Dong Wang, Jian Li, Hao Zhou, Lei Li, and Yitan Li. BRITS: Bidirectional Recurrent Imputation for Time Series. *Advances in neural information processing systems*, 31, 2018.
>
> [2] Wenjie Du, David Cot́e, and Yan Liu. Saits: Self-attention-based imputation for time series. *Expert Systems with Applications*, 219:119619, 2023.
>
> [3] Satya Narayan Shukla and Benjamin Marlin. Multi-time attention networks for irregularly sampled time series. In *International Conference on Learning Representations*, 2021.
>
> [4] Yusuke Tashiro, Jiaming Song, Yang Song, and Stefano Ermon. Csdi: Conditional score-based diffusion models for probabilistic time series imputation. *Advances in Neural Information Processing Systems*, 34:24804–24816, 2021
>
> [5] Rahul G Krishnan, Uri Shalit, and David Sontag. Deep Kalman Filters. *arXiv* preprint
> arXiv:1511.05121, 2015.

---

> ### Comment · Reviewer_wRxf · 2023-11-17
> **opinions**
>
> 1. I agree with reviewer yyLt that more recent baselines should be considered. In the updated manuscript, the authors have updated the results from several strong baselines and the proposed method is better or comparable to these baselines.
>
> 2. I agree that we need some metric as a piece of evidence to show the extent of regularization. Unfortunately, it is currently missing from this paper. However empirical results do show that CIB is better than IB methods, which partly support the claim made in this paper.

---

> ### Author Response · Authors · 2023-11-20
> **Dear Reviewer yyLt**
>
> Once again, we would like to thank you for your invaluable feedback! We were wondering whether our response from November 16 has sufficiently addressed your questions. Also we thank to Reviewer wRxf on the comment. If you have any leftover comments or questions, please let us know - we would be happy to do our utmost to address them!
>
> Paper 4032 Authors

---

> ### Comment · Reviewer_Vm5N · 2023-11-20
> **Agree with Reviewer wRxf and yyLt**
>
> I would like to point out that I also agree with point 2 from reviewers yyLy and wRxf: I also believe a proper metric showing the effective regularization from the proposed CIB is missing.

---

> ### Author Response · Authors · 2023-11-21
> **More on quantitative assessment of regularization on preserving temporal information.**
>
> We appreciate Reviewer yyLt, wRxf, and Vm5N on the comment.
>
> In addition to the qualitative assessment on $I(Z_t;X^o_{\setminus t})$ shown in Figure 1 and empirical performances in Section 5, we compare the approximated quantity on $I(Z_t;X^o_{\setminus t})$ using (9) to assert that regularization term in our CIB objective indeed preserves more temporal information. Please note that, as derived in (A.4), the lower bound is given by $I(Z_t;X^o_{\setminus t}) \ge \log(N) - L^3_\phi$, which depends on the batch size $N$ and the objective $L^3_\phi$. Table below shows the values of  $L^3_\phi$, indicating that a lower value corresponds to more preserved information.
>
> | Dataset  | VAE | HI-VAE | GP-VAE | Ours (Uniform) | Ours (Kernel) |
> |--|--|--|--|--|--|
> | RotatedMNIST | $3.619 \pm 0.005$ | $3.577 \pm 0.008$ | $3.435 \pm 0.001$ | $\mathbf{3.428} \pm 0.006$ | $\mathbf{3.418} \pm 0.002$ |
> | Beijing | $3.950 \pm 0.015$ | $3.822 \pm 0.012$ | $3.762 \pm 0.012$ | $\mathbf{3.700} \pm 0.008$ | $\mathbf{3.686} \pm 0.006$ |
>
>
> This results demonstrates that regularization in our CIB objective preserves more information in $I(Z_t;X^o_{\setminus t})$ based on the approximation. We also added the Appendix B.3 in the revised manuscript, illustrating above experiments and results.
>
> Again, thank you for the feedback and we look forward to further discussions if you have additional questions or suggestions!

---

> > ### Comment · Reviewer_yyLt · 2023-11-22
> >
> > Thanks for the authors' detailed rebuttal. After I have read all the reviewers' comments and the responses, I think most of my concerns are well addressed. I'd like to raise my score from 5 to 6.

---

### Official Review · Reviewer_mVWF · 2023-10-31

**Soundness:** 3 good
**Presentation:** 2 fair
**Contribution:** 3 good
**Rating:** 6
**Confidence:** 4

**Summary:**

This paper focuses on time series imputation. The authors believe that information bottleneck (IB) is a well-suited theoretical foundation for imputation task. While directly applying conventional IB framework may results in substantial loss of temporal dependencies, the authors propose a novel conditional information bottleneck (CIB) approach. The proposed CIB suggests the model to learn more information about latent representation from temporal contexts. The idea is novel and interesting, which is supported by some theoretical analysis and experimental result.

**Strengths:**

1. The proposed TimeCIB is novel and technical sound.
2. The theoretical analysis is solid, making the whole framework more persuasive and reliable.
3. Smoothing the latent representation is interesting and novel.

**Weaknesses:**

1. The motivation of using conditional term $X^o_t | X^o_{\\t}$ should be further explained. Why using the conditional term can tackle the issue of conventional IB?
2. A figure of the whole framework will be helpful for understanding this work.
3. The compared baselines are out of date. There exist massive works on time series imputation, the authors should compare their method with more recent works.
4. It would be appreciated if the authors could provide code.

**Questions:**

See weaknesses.

---

> ### Author Response · Authors · 2023-11-16
> **Response to Reviewer mVWF.**
>
> We thank the reviewer for your thoughtful comments and suggestions. We appreciate your recognition of our problem setting and approach. Below, we provide responses to each of your comments in turn.
>
> ### **[W1] Motivation of the conditional term**
>
> Please allow us to explain the motivation behind the conditional term using a compelling example illustrated in Figure 1.
>
> The conventional IB comprises two terms: one reconstruction and one regularization. As illustrated in Figure 1A left, a simple extension of the conventional IB to time series imputation aims at maximizing the information related to $X_t$ (red) while minimizing the information from the entire time series input $X^{o}_{1:T}$ (blue). However, such regularization can be overly strong, impeding the model's ability to accurately impute missing values by capturing the underlying temporal dynamics. As depicted in Figure 1B, the conventional IB may fail in interpolation or extrapolation scenarios where the model needs to gather information about the temporal dynamics
>
> We tackle the problem of \textit{loss of temporal dynamics} by modifying the regularization term in the IB framework. Specifically, we propose CIB to preserve information from the remaining timesteps. As illustrated in Figure 1A right, the CIB does not minimize the information from $X_{\setminus t}^o$, when maximizing information that is required for reconstructing $X_t$. This is mathematically confirmed by (5), where applying the chain rule gives that the conditional regularization encourages conciseness by minimizing $I(Z_{t}; X^o_{1:T}$), while capturing the underlying temporal dynamics by maximizing $I(Z_{t}; X^{o}_{\setminus t}$).
>
>
> ### **[W2] A schematic illustration of our framework**
>
> We have added a figure illustrating our framework in Appendix C (page 19).  We hope this helps readers to better understand our work.
>
>
> ### **[W3] Additional cutting-edge benchmarks**
>
> We have included the following **four** cutting-edge deep learning-based time series imputation  methods: two predictive methods including an RNN-based method, BRITS [1], and a transformer-based methods, SAITS [2], and two generative methods including an autoencoder-based method, mTANs [3], and a diffusion-based method, CSDI [4].
>
> Due to the limited rebuttal period, we have provided results for the Beijing, USLocal, and Physionet2012 datasets in the revised manuscript. Please find the result table on the general response and the revised manuscript. To summarize, our method achieves the best performance on the USLocal dataset, which has the longest time steps (T=168) and clear periodicity. For the Beijing dataset, our method performs slightly better than or comparably to state-of-the-art predictive methods. Here, CSDI (with n_samples=25) demonstrates the best performance but with a computation time approximately 2000 times slower than ours due to the required sampling process. For Physionet dataset, our method outperforms the cutting-edge benchmarks in terms of AUROC suggesting that our approach recovers semantically meaningful information for downstream tasks.
>
>
> ### **[W4] Source code**
>
> We will upload the entire source code on Githhub upon acceptance.
>
> [1] Wei Cao, Dong Wang, Jian Li, Hao Zhou, Lei Li, and Yitan Li. BRITS: Bidirectional Recurrent Imputation for Time Series. *Advances in neural information processing systems*, 31, 2018.
>
> [2] Wenjie Du, David Cot́e, and Yan Liu. Saits: Self-attention-based imputation for time series. *Expert Systems with Applications*, 219:119619, 2023.
>
> [3] Satya Narayan Shukla and Benjamin Marlin. Multi-time attention networks for irregularly sampled time series. In *International Conference on Learning Representations*, 2021.
>
> [4] Yusuke Tashiro, Jiaming Song, Yang Song, and Stefano Ermon. Csdi: Conditional score-based diffusion models for probabilistic time series imputation. *Advances in Neural Information Processing Systems*, 34:24804–24816, 2021

---

> > ### Comment · Reviewer_mVWF · 2023-11-21
> >
> > Thanks for the authors' detailed rebuttal. Most of my concerns are well addressed, but to me, the motivation of the conditional term still lacks intuitive explanation or theoretical basis. So I will keep my positive score as it is.

---

> ### Author Response · Authors · 2023-11-20
> **Dear Reviewer mVWF**
>
> Once again, we would like to thank you for your invaluable feedback! We were wondering whether our response from November 16 has sufficiently addressed your questions. If you have any leftover comments or questions, please let us know - we would be happy to do our utmost to address them!
>
> Paper 4032 Authors

---

### Official Review · Reviewer_Vm5N · 2023-10-31

**Soundness:** 3 good
**Presentation:** 4 excellent
**Contribution:** 3 good
**Rating:** 6
**Confidence:** 4

**Summary:**

The paper *Cconditional Information Bottleneck Approach for Time Series* extends the information bottleneck (IB) to deal with time series imputation by proposing a novel conditional information bottleneck. The authors also propose a novel deep learning method that can approximately achieve the proposed CIB objective for time series imputation using the lower bound and a novel temporal kernel. The proposed model is compared against some state-of-the-art VAE-based models for time series imputations under three different datasets, showing promising results and outperforming the other baselines.

**Strengths:**

The paper is well-structured and easy to follow. The motivation and the proposed method are very well described in the manuscript and nicely derived in the Appendix. The more tricky points are also well described (e.g. the application of the chain rule in Equation 5). The related work section is also nicely depicted, making clear that the authors are aware of the most recent works for missing data imputations in temporal series. Finally, the results are concise and clear to read. It is evident from the results that their proposed method opens very prospective lines for future research in missing data imputation in time series.

Furthermore, the paper holds interest from an information bottleneck perspective in two significant ways. Firstly, it extends the information bottleneck concept to a conditional information bottleneck, representing a novel contribution in its own right. Secondly, the introduction of a fresh objective for VAE-based models to address missing data concerns is noteworthy. The authors follow the standard assumptions for handling missing data using VAEs, such as evaluating the objective on the observations of filling the missing points with zeros. However, the paper sets itself apart not solely due to its new objective based on the conditional information bottleneck but also because of its capacity to generate fresh avenues of research and stimulate discussions during the conference

**Weaknesses:**

I would like to point out some weaknesses that I consider the paper exhibits and that could be further analyzed by the authors:

1. As many papers dealing with temporal data, sometimes notation becomes a bit complex and makes the reading more difficult.
2. I believe more information about the experimental section could be provided. That, is, for HealingMNIST and RotatedMNIST, which kind of CNN-based encoder/decoder architecture is being used; for Physionet, which kind of networks are used? There is no information about this, especially for the proposed method, since for baselines this can be accessed in their referred papers. And this is quite important when asking why the proposed method outperforms any other baseline. This leads me to the following point.
3. I am a bit hesitant about the superior performance of the proposed method in (almost) every dataset and performance metric. This creates some doubts that will be asked in the question section.
4. There is no reference for code availability to reproduce the experiments. Will you upload the code to Git Hub or any available repository? This is rather important for the sake of reproducibility. And given that the results are (fairly always) better than baselines, this should be a must-have for this paper.
5. I find the paper rather novel in terms of the proposal of the CIB objective. However, in terms of the VAE-based model, I believe there is much inspiration coming from GPVAE [1], due to the idea of including an extra kernel to further enhance the learning of the temporal dynamics. Is this the case? Should the authors be more worried about properly referencing this work?
6. An ablation study should be included to determine the importance of each contribution in the paper. That is, i) how the different terms in the proposed objective influence the optimization ii) how important is it to introduce the kernel — without the kernel, would the results approach the ones from GPVAE, or does the proposed model still outperform other baselines? I think it is important to analyze each contribution of the paper to obtain a clear and detailed view of the behaviour of the proposed model.

[1] Fortuin, V., Baranchuk, D., Rätsch, G., & Mandt, S. (2020, June). Gp-vae: Deep probabilistic time series imputation. In *International conference on artificial intelligence and statistics* (pp. 1651-1661). PMLR.

**Questions:**

Here are some questions that I would like the authors to address:

1. In Equation 7, in the denominator, why do you have a summation over X when you don’t have any variable x, just z? This also holds for Equation 9. I believe you mean that you sample from the encoder to get z out of x. However, I think the equation could be improved.
2. The samples from Equation 4, how are they taken? How many samples? Is the method sensible to the number of samples? Do the results improve by taking more samples? Did you consider the following approaches such as MIWAE?
3. Why do you always outperform all baselines, in all datasets under any missing data assumption? I find it rather surprising given that you have a loss (Equation 8) composed of three terms, which I believe must be difficult to optimize. Besides, there is not much information about the $\beta$ and $\gamma$ parameters. How do they influence the optimization? There is no discussion about this.
4. What do black crosses and black dots represent in Figure 3? I believe dots are observations and crosses missing points, but this is not commented on.
5. Looking at Figure 3a), for example: how is it possible that HIVAE and GP-VAE obtain such poor results compared to your proposed method? I assume this is the case for HIVAE since it is not designed to deal with temporal data, but for GP-VAE this is very surprising. As described in [2], I would expect this to happen in the missing parts, where the GPVAE would produce a more “mean” solution less correlated to the original signal. However, in the observations, I would expect the GP-VAE to obtain better results.
6. In Table 3 you have an interesting result. In terms of RMSE, it can be observed that mean imputation (one of the most naive approaches) outperforms any baseline, even state-of-the-art methods. What do you think about this result? As observed in [2], sometimes error metrics are inconclusive when evaluating temporal scenarios. Did you find this behaviour in other datasets? Did you think about the possibility of instead of using missing data randomly or following other mechanisms (which you withdrew from GPVAE and other papers I believe) using missing data in burts or sequences, where actually it is the standard scenario you would find in a temporal scenario such as in healthcare? This could be an interesting point to be analyzed here: check whether the solutions from baselines and proposed methods are more correlated.

[1] Mattei, P. A., & Frellsen, J. (2019, May). MIWAE: Deep generative modelling and imputation of incomplete data sets. In *International conference on machine learning* (pp. 4413-4423). PMLR.

[2] Barrejón, D., Olmos, P. M., & Artés-Rodríguez, A. (2021). Medical data wrangling with sequential variational autoencoders. *IEEE Journal of Biomedical and Health Informatics*, *26*(6), 2737-2745.

---

> ### Author Response · Authors · 2023-11-16
> **Response to Reviewer Vm5N (part 1/2).**
>
> We thank the reviewer for your thoughtful comments and suggestions. We appreciate your recognition of our problem setting and approach. Below, we provide responses to each of your comments and questions in turn.
>
> ### **[W2] Architecture specification**
> In our original submission, we specified architecture specifications in Appendix C.1 due to the page limit. We have added a guiding sentence in the revised manuscript.
>
> Here, we briefly summarize implementation details. To highlight the contribution of our novel objective, our implementation adopts most of the settings used in GP-VAE [1]. For the HealingMNIST and RotatedMNIST datasets, we employ one layer of a CNN with a kernel size of 3, followed by a stochastic encoder consisting of two-layer LSTM. The decoder is a two-layer feed-forward network. The primary architectural distinction of our method from GP-VAE is the stochastic encoder; GP-VAE utilizes dimension-wise encoding to employ a Gaussian process prior, while our method utilizes latent variables at each time step similar to conventional VAEs.
>
> ### **[W5] Relation to GP-VAE**
> We appreciate the reviewer for bringing this important point to our attention and apologize for the unclear comparison with GP-VAE. As you mentioned, we admit that our method and GP-VAE share some philosophy on how to tackle time series imputation and use temporal kernels. However, there are key differences between our work and GP-VAE. Under the IB framework, GP-VAE models the smooth temporal evolution of latent variables by replacing the conventional unit Gaussian prior with a GP prior specified by temporal kernels.
> However, our work is motivated by the inherent limitation of the IB in discarding temporal information (as discussed in Section 3.1) and proposes a novel CIB principle that alleviates the strict regularization of IB. In our approach, temporal kernels are **optionally** adopted to introduce an inductive bias about the underlying temporal dynamics. This is different from GP-VAE, where the temporal kernel is introduced specifically for the GP prior.
>
> We have included this detailed discussion about the relation to GP-VAE in the Related Work of the revised manuscript (page 7).
>
> ### **[W6 (ii)] Regarding the introduction of temporal kernel**
>
> As mentioned in (Relation to GP-VAE), we optionally adopt temporal kernels to introduce an inductive bias about the underlying temporal dynamics. So, the main novelty of our work comes from how we model the time series imputation using the conditional IB framework, rather than introducing the temporal kernel. The imputation performance of our method without kernel (denoted as uniform) still demonstrates significant improvement, supporting our claim.
>
> ### **[Q3, W6 (i)] Discussions on beta and gamma - Sensitivity analysis and optimization**
>
> As the reviewer suggested, we have included sensitivity analysis and optimization strategy  on beta and gamma. Before presenting our new results, please allow us to redefine optimization hyperparameters as the form of $\min L^1_{\phi, \theta} + \beta’ L_\phi^2 + \gamma’ L_\phi^3$, so that we can isolate effects of each regularization terms given fixed reconstruction. (Please refer to Appendix B3 for details)
>
> **Sensitivity on $\beta’$.** Manipulating $\beta’$ is related to the regularization trade-off that a low $\beta’$ (weak regularization) can cause a lack of time series-wise information, which may result in overfitting solely on observed values while a high $\beta’$ (strong regularization) can cause loss of timepoint-wise information since all distribution will become equivalent to unit Gaussian. We have included a qualitative analysis on HealingMNIST, showing these behaviors in Figure B2(a) in in Appendix B3 (page18).
>
> **Sensitivity on $\gamma’$.** Since $\gamma’$ basically works as a regularization term, sensitivity on $\gamma’$ is similar to $\beta’$, but the behaviour is slightly different. For a low $\gamma’$, there is no driving force on conserving temporal dynamics (see Section 3.1), thereby causing a loss of temporal dynamics and would eventually become equivalent to HI-VAE when $\gamma’=0$. A high $\gamma’$ also can cause a loss of timepoint-wise information, since $\mathcal{L}_3$ will map all $z_t$ from the same time series into a single point (achieving maximum alignment), which means that we cannot distinguish individual timepoints. We have included a qualitative analysis on HealingMNIST, showing these behaviors in Figure B2(b) in in Appendix B3 (page18).
>
> **Optimization Strategy.** It is true that simultaneous optimization of $\beta$ and $\gamma$ could be difficult. However, based on the above sensitivity analysis, we provide a practical suggestion: First, we set $\gamma’=0$ and find optimal $beta’$ (which is HI-VAE). Second, under a similar range of order of magnitude of $\beta’$, find the optimal $\gamma’$.
>
> We have included above discussions in Appendix B3 of the revised version.

---

> ### Author Response · Authors · 2023-11-16
> **Response to Reviewer Vm5N (part 2/2).**
>
> ### **[Q2] Discussions on sampling**
>
> **Implementation of Sampling.** In our experiments, we implement (4) with exactly the same way as the traditional VAE methods, and take one sample. One can draw many samples to achieve stable approximation, but we couldn’t find significant improvements in our experiments.
>
> **Discussion about MIWAE.** MIWAE [2] is an extension of IWAE [3] which proposes an importance-weighted autoencoder for imputation under the ‘missing-at-random (MAR)’ assumption. Since their objective (Equation 4 of [2]) does not originate from the information theory, we are uncertain about how we can incorporate this into our information-theoretic framework. (Please note that VAE is a special case of the IB framework with $\beta = 1$.) Despite this, a subset of our objective (i.e., $L^1$ and $L^2$) share a similar mathematical form with traditional ELBO, which allows us to run experiments by replacing $\beta L^1 + L^2$ term (after setting $\beta=1$) with $L^{MIWAE}$ with k=5, 10 on HealingMNIST dataset. Combination with importance sampling used in MIWAE showed improved MSE when applied to VAE, but we could not observe performance gain when applied to ours (MSE = $0.226 \pm 0.000$ for K=5, and $0.225 \pm 0.001$ for K=10).  We claim that such results come from two reasons: (i) as emphasized in [2], the objective in [2] assumes the MAR assumption, while HealingMNIST (and most of the real-world time series) does not satisfy this (we will revisit this in the next response), and (ii) connection between [2] and information-theoretic approach is not solid.
>
> Nevertheless, we admit that a reliable sampling strategy is crucial for probabilistic imputation models. Although this is out of the scope of our paper, addressing (i) or (ii) could be interesting future works.
>
> ### **[Q6] Mean imputation results in Table 3**
>
> We appreciate your valuable insight and suggestions. Inspired by [4], we conducted an extensive input/output data analysis and observed an interesting discussion topic about outliers. This result suggests intriguing future research topics on latent state transition.
>
> To identify the source of failure in the MSE metric, we have analyzed the distribution of input data and the MSE of our model outputs. Interestingly, it turns out that the majority of error metric comes from very few data points, where the ground truth values of the missing data point are outliers (after removing 0.1 % of the highest errors, our model achieves MSE=$0.409$). Generally speaking, this could be interpreted as an outlier problem — whether the model should smooth out outliers or respect all data points. As discussed in [4], deep learning methods in general smooth out the outliers, and especially when we use temporal kernels, this error becomes larger.
>
> We questioned why this especially happens in Physionet2012. Is this simply because of the huge missing ratio? We found a counter-example to the previous claim in our robustness experiments on missing ratio (Figure 2(b) and Table B4), where deep learning models still performed better than mean/forward imputations, even when missing ratio = 0.9.
>
> After a careful analysis of outliers, we hypothesized that the combination of i) latent state change and ii) a high missing ratio corrupts MSE-based evaluation. We observed that the explosion of MSE occurs when missingness happens after a drastic change in patients' status. If a model sees only one observation after the status shift (which appears as an outlier from the observation) and predicts several observations afterward, the model would smooth the effect of the outlier. The fact that [4] mitigates this problem by introducing Gaussian mixture latent variables also supports this hypothesis.
>
> The above observation suggests an intriguing problem regarding modeling latent states from incomplete time series. To address this issue, the model should solve two problems simultaneously: time series imputation and latent state prediction. Fortunately, these two problems seem to be related to each other—good predictions on latent states enhance imputation, and good imputation helps state prediction. Since our objective involves contrastive learning, one could expand it by adopting ideas from contrastive partial label learning [5]. As this topic is beyond the scope of this paper, we will leave it for future work.
>
> ### **We also corrected the comments below:**
>
> [W1] We provide a notation table in Appendix E.
>
> [W4] We will upload the whole source code to Github.
>
> [Q1] We improved (7) and (9).
>
> [Q4] You are correct that dots are observed, and crosses are withdrawn values. We have revised the caption.
>
> [Q5] This happens by the sampling, and GP-VAE often make mistakes in capturing the mean of fluctuation (see Figure 6 and 7 of [4]).  We will provide more examples showing more qualitative results in the final manuscript.

---

> ### Author Response · Authors · 2023-11-16
> **References**
>
> [1] Fortuin, V., Baranchuk, D., Rätsch, G., & Mandt, S. (2020, June). Gp-vae: Deep probabilistic time series imputation. In *International conference on artificial intelligence and statistics* (pp. 1651-1661). PMLR.
>
> [2] Mattei, P. A., & Frellsen, J. (2019, May). MIWAE: Deep generative modelling and imputation of incomplete data sets. In *International conference on machine learning* (pp. 4413-4423). PMLR.
>
> [3] Burda, Y., Grosse, R., Salakhutdinov, R. (2016). Importance Weighted Autoencoders. In *International converence on Learning Representations*
>
> [4] Barrejón, D., Olmos, P. M., & Artés-Rodríguez, A. (2021). Medical data wrangling with sequential variational autoencoders. *IEEE Journal of Biomedical and Health Informatics*, *26*(6), 2737-2745.
>
> [5] Wang et. al. (2023). PiCO: Contrastive Label Disambiguation for Partial Label Learning. In *International converence on Learning Representations*

---

> ### Author Response · Authors · 2023-11-20
> **Dear Reviewer Vm5N**
>
> Once again, we would like to thank you for your invaluable feedback! We were wondering whether our response from November 16 has sufficiently addressed your questions. If you have any leftover comments or questions, please let us know - we would be happy to do our utmost to address them!
>
> Paper 4032 Authors

---

### Author Response · Authors · 2023-11-16
**General Response**

Dear reviewers and AC,

We sincerely thank all the reviewers and AC for your enormous effort and time spent reviewing our manuscript.
We would share some additional results with four additional state-of-the-art time-series imputation methods. Please see the table below (also edited in our revised manuscript).

Also, in our revised manuscripts, the revised contents are highlighted in $\textcolor{blue}{blue}$.

We hope that the updates can answer to the questions and help clarify the effectiveness of our method.

Thank you very much!
Authors
Best,

Authors.

$\quad$
$\quad$

### **Additional Experiments**

|Methods|Beijing| | | US Local  ||| Physionet | | |
|----|-----|-----|------|-----|------|-----|-----|------|-----|
| | NLL |MSE|ForecastMSE| NLL |MSE | ForecastMSE | NLL |MSE | AUROC |
|BRITS     | - | $0.396$ | $0.490 $| - | $0.384$ | $0.398$| - | $0.529$ | $0.700$|
|SAITS     | -| $\mathbf{0.283}$ | $0.450 $| - | $0.275$ | $0.350 $| - | $0.501$ | $0.713$|
|mTANs    | *| $0.287$ | $0.436 $| * | $0.268$ | $\mathbf{0.337}$| * | $\mathbf{0.499}$ | $0.721$|
|CSDI(n=5) | * | $0.287$ | $\mathbf{0.423} $| * | $0.378$ | $0.364 $| * | $0.548$ | $0.705$|
|CSDI(n=25) | * | $\mathbf{0.270}$ | $\mathbf{0.423} $| * | $0.340$ | $0.347 $| * | $\mathbf{0.478}$ | $0.683$|
| VAE        | $1.427$| $1.016$ | $0.524 $| $1.462$ | $1.086$ | $0.467$| $1.400$ | $0.384$ | $0.691$|
| HI-VAE   | $1.081$| $0.321$ | $0.464 $| $1.078$ | $0.317$ | $0.380 $| $1.345$ | $0.384$ | $0.696$|
| GP-VAE  | $1.077$| $0.316$ | $0.463 $| $1.078$ | $0.318$ | $0.385$| $1.227$ | $0.384$ | $0.730$|
|----------|---------|---------|----------|---------|----------|---------|---------|--------|---------|
|Ours(uniform) |$\mathbf{1.063}$| $0.291$ | $0.445 $| $\mathbf{1.052}$ | $\mathbf{0.265}$ | $0.351$| $\mathbf{1.183}$ | $0.528$ | $\mathbf{0.744}$|
|Ours(Periodic) |$\mathbf{1.060}$| $\mathbf{0.283}$ | $0.443$| $\mathbf{1.049}$ | $\mathbf{0.260}$ | $\mathbf{0.327}$| $\mathbf{1.179}$ | $0.521$ | $\mathbf{0.744}$|

We bolded two best results of each metric. For a fair comparison, the magnitudes of the number of parameters are set to be equivalent among the evaluated methods (added at Table C2, page 20). Asterisks (*) means that models are probabilistic but cannot compare NLL (please see footnotes in page 7).

---

### Public Comment · ~Gehua_Ma1 · 2023-11-22
**Suggesting related work**

This work presents a novel conditional IB framework for time series. And we're glad to note that we've previously adopted a quite similar idea in our previous work [1]. Namely, using conditional information bottleneck for time series prediction (neural data).
However, the CIB method introduced here exhibits greater generality and is expected to be highly effective in a much wider range of time series tasks. Congrats on the awesome work.

[1] G. Ma et al. Temporal Conditioning Spiking Latent Variable Models of the Neural Response to Natural Visual Scenes. NeurIPS 2023.  https://openreview.net/forum?id=V4YeOvsQfu

Best, Marcus

---

### Meta-Review · Area_Chair_xKz7 · 2023-12-08

**Metareview:**

Although there were several critical remarks (regarding, for instance, the evaluation metric used), finally all reviewers agreed that this paper contains some novel aspects, that it addresses a relevant problem and that the theoretical contributions are interesting. In general, I share this positive over-all impression, and therefore, I recommend acceptance of this paper.

**Justification For Why Not Higher Score:**

There are still some open questions regarding the evaluation metric used.

**Justification For Why Not Lower Score:**

Interesting and novel theoretical results.

---

### Decision · Program_Chairs · 2024-01-16

Accept (poster)